# 🪄 Lumos: Language Agents with Unified Formats, Modular Design, and Open-Source LLMs

## Abstract

In this paper, we present 🪄 Lumos, **L**anguage agents with **U**nified formats, **M**odular design, and **O**pen **S**ource LLMs. Lumos features a modular architecture consisting of planning, grounding, and execution modules built based on open-source LLMs such as LLAMA-2. The *planning module* decomposes a task into a sequence of high-level subgoals; the *grounding module* then grounds the generated subgoals to a series of low-level actions that can then be executed by the *execution module*. To obtain high-quality annotations for training these modules, we leverage LLMs to convert ground-truth intermediate reasoning steps in existing benchmarks into a unified format that can be used in the Lumos framework. Lumos achieves competitive or superior performance compared to the state of the art on a variety of complex interactive tasks. We observe: (1) Lumos is competitive with the LLM agents that are $2 - 4\times$ larger on maths tasks, and outperforms GPT-4/3.5-based agents on complex QA and web agent tasks; (2) Lumos showd superior performance against open-source agent baseline formulations including chain-of-thoughts fine-tuning and unmodularized training; (3) Lumos surpasses larger LLM-based agents on an unseen interactive task, WebShop, and achieves 5-10 reward improvement over domain-specific agents.

## 1 Introduction

Language agents (Yao et al., 2022b; Patil et al., 2023; Xu et al., 2023a; Lu et al., 2023; Liu et al., 2023b) harness knowledge learned from language to solve complex problems by generating sequences of coherent actions. As a result, they have become a critical component of systems that solve complex interactive tasks, such as maths (Cobbe et al., 2021; Patel et al., 2021), complex question-answering (Yang et al., 2018; Geva et al., 2021), and web agent tasks that ask agents to operate on websites to accomplish user requests (Deng et al., 2023; Zhou et al., 2023). Solving such tasks typically requires long-horizon planning, as well as interaction with tools and environments.

Prior works (Yao et al., 2022b; Shinn et al., 2023; Paranjape et al., 2023; Xu et al., 2023a) mainly rely on prompting closed API-based Large Language Models (LLMs)[1] such as OpenAI GPT-4/3.5 (OpenAI, 2023; 2022) to instruct them how to decompose tasks and actively respond to dynamic environments. Recently, there have also emerged open-source LLM-based agents such as WizardLM-30B (Xu et al., 2023b) and ReWOO-Planner-7B (Xu et al., 2023a). However, Liu et al. (2023a); Xu et al. (2023a) show that their performance largely lags behind their closed counterparts. This suggests that a sufficiently large LLM (e.g., approximately 175B-scale GPT-3/3.5/4) is a prerequisite of developing an effective language agent.

*Can language agents with orders of magnitude smaller size 7B be as effective?* To this end, we propose Lumos, **L**anguage agents with **U**nified formats, **M**odular design, and **O**pen **S**ource LLMs. The architecture of Lumos is designed with modularity in mind, featuring planning, grounding, and execution modules that are constructed using smaller open-source LLMs such as LLAMA-2-7B (Touvron et al., 2023). The planning module dissects a complex task into a sequence of high-level subgoals, and the grounding module subsequently translates these generated subgoals into a series of low-level actions, which can be executed by the execution module. We propose two Lumos formulations, Lumos-Onetime (Lumos-O) and Lumos-Iterative (Lumos-I), to establish the interaction

---

[1]We refer to models whose weights have not been publicly released as of September 28, 2023 as "closed".

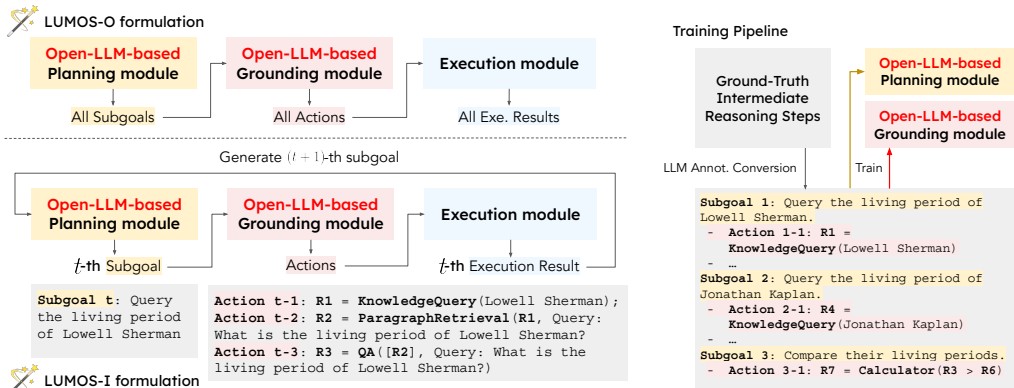

(a) Overview of our proposed Lumos formulations – Lumos-Onetime (Lumos-O) and Lumos-Iterative (Lumos-I) (see details in §2).

(b) The process of acquiring annotations for training planning and grounding modules (see details in §3).

Figure 1: Overall architecture, formulations and training pipeline of Lumos.

between different modules. Demonstrated in Figure 1a, Lumos-O is an efficient formulation that generates all the subgoals and executable actions at once using a one-time inference call. Lumos-I is an adaptive formulation that iteratively generates one subgoal and its corresponding executable actions according to the current environment states.

Fine-tuning the modules with ground-truth data for planning and grounding would enable language agents to acquire these skills (Wang et al., 2023b; Shridhar et al., 2023; Li et al., 2023). However, a realistic challenge is that there are no datasets containing these types of annotations — that is high-level subgoals (e.g., "*Subgoal 1: Query the living period of Lowell Sherman*") and executable actions (e.g., KnowledgeQuery(Lowell Sherman)). To obtain high-quality annotations for training the modules, as shown in Figure 1b, we harness LLMs to convert ground-truth intermediate reasoning steps in existing benchmarks into a unified annotation format compatible with the Lumos framework. This approach not only aids in obtaining high-quality training annotations as the conversion is based upon high-quality ground-truth reasoning steps in existing datasets, but the unified format also facilitates the training of modules with enhanced generalizability across various domains of complex interactive tasks, as shown by our experiments.

Lumos demonstrates competitive or state-of-the-art performance on various complex interactive tasks, including maths, complex QA and web agent tasks. Specifically: (1) On maths tasks GSM8K (Cobbe et al., 2021) and SVAMP (Patel et al., 2021), Lumos competes effectively with agents that are $2-4\times$ larger in size, and it even surpasses GPT-based agent frameworks in complex QA (Geva et al., 2021; Yang et al., 2018) and web agent tasks (Deng et al., 2023). In particular, Lumos achieves 3.9-8.5% LLM accuracy gain over GPT-3.5-turbo-based agents on HotpotQA, as well as 5.1% step success rate gain over GPT-4 on Mind2Web. (2) Compared to other open-source agent baseline formulations such as chain-of-thoughts fine-tuning and unmodularized agent training, Lumos formulations consistently exhibit superior performance. (3) On an unseen interactive task called WebShop (Yao et al., 2022a), Lumos outperforms larger LLM-based agents (e.g., WizardLM-30B) around 20 reward improvement and delivers a remarkable 5-10 reward improvement over domain-specific agents. Together, our proposed framework and unified model provides valuable resource and direction for advancing open-source interactive language agents.

## 2 ✨ Lumos: A Modular Agent Framework

We introduce the overall design and two formulations for developing agents within this framework.

### 2.1 Lumos Agent Architecture

To solve a complex interactive task, it is essential to first decompose the task to a sequence of subgoals (i.e., planning), then convert each subgoal to a sequence of executable actions (i.e., grounding),

and finally execute the actions. Therefore, there are three modules in LUMOS for planning, grounding, and execution, respectively.

**Planning Module (PM).** The module is designed to decompose a complex task into a series of high-level subgoals, which are written in natural language. For example, there are three subgoals to answer a question "*Who lived longer, Lowell Sherman or Jonathan Kaplan?*", as shown in Figure 1b: (1) Query the living period of 'Lowell Sherman'; (2) Query the living period of 'Jonathan Kaplan'; (3) Compare their living periods and return the answer. The generated subgoals can help agents translate a complex task to low-level actions in an interpretable manner. Note that the subgoals are not executable and not bound to any specific tools or APIs.

**Grounding Module (GM).** This module aims at converting the high-level subgoals produced by the PM to low-level executable *actions*. To accomplish the subgoal "*Query the living period of Lowell Sherman*", the GM converts it to one or multiple actions, e.g., KnowledgeQuery(Lowell Sherman), QA([R2], Query: *"What is the living period of Lowell Sherman?"*) that query the information needed for solving the given subgoal.

**Execution Module (EM).** The execution module is a program that parses actions to a series of external tools including APIs, small neural models, and virtual simulators that interact with relevant tools and external environment. For example, the action type KnowledgeQuery can be linked to the Wikipedia API of querying a document. The web agent tasks also require the interaction with raw HTML via a simulator that supports website operations such as typing, clicking and selecting.

The main characteristics of LUMOS framework is the interaction between planning, grounding and execution modules. We propose the following two formulations that foster the communication between the three modules: LUMOS-Onetime (LUMOS-O) and LUMOS-Iterative (LUMOS-I).

## 2.2 LUMOS-ONETIME (LUMOS-O)

LUMOS-Onetime (LUMOS-O) is an efficient formulation that generates all the subgoals and grounded actions at once. In the LUMOS-O formulation described in Figure 1a, the planning module generates all the three subgoals with a single inference and no interaction with the other two modules. We pass all generated subgoals altogether into the grounding module that converts them to a chain of low-level actions at once without interacting with execution modules. Given the task and three subgoals, the grounding module would directly compute all converted low-level actions such as KnowledgeQuery(Lowell Sherman) and ParagraphRetrieve(..., Query: "*How long is the living period of Lowell Sherman?*").

Besides the task description and subgoals to be grounded, action interfaces are also an indispensable part of grounding module's input. The action interfaces guide the grounding module to produce valid executable actions. For example, we provide the action interface "SolveEquation(equation): Solve the previous set equation" to the grounding module. It defines the action name "SolveEquation" and action functionality "Solve the previous set equation". It also forces acceptable action arguments to be an equation.

More formally, we illustrate the overall planning and grounding process of LUMOS-O in Figure 2a. For planning, given the parameters of planning module is $\theta_{plan}$, we input the task description $T$ into the planning module. The generated output would be a series of subgoals, $S = \theta_{plan}(T) = \{s_1, ..., s_n\}$. Since grounding relies on the task description, action interface $I = \{i_1, ..., i_k\}$ and subgoals generated by the planning module, the grounded actions could be obtained via $A = \theta_{ground}(T, I, S)$, where $\theta_{ground}$ represents the grounding module parameters.

## 2.3 LUMOS-ITERATIVE (LUMOS-I)

LUMOS-Iterative (LUMOS-I) is a formulation that generates one subgoal and its corresponding executable actions in each iteration. When generating the current $t$-th subgoal, the planning module requires the previous planned subgoals and the execution results of their grounded actions. The execution results assist the planning module to be aware of the dynamic environmental change and thus decide actions according to up-to-date environments. We take complex QA task as example to show its advantage of flexibly adapting to the environmental change. Suppose that we are answering the question "*Which state was the U.S. president in 2014 born in?*".

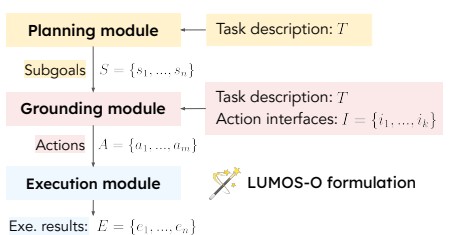 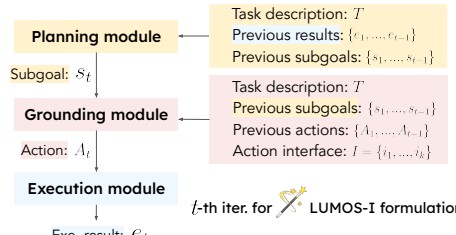

(a) Inputs & outputs of each module in LUMOS-O formulation.

(b) Inputs & outputs of each module for the $t$-th iteration in LUMOS-I formulation.

Figure 2: Overview of the inputs & outputs contents for the modules in LUMOS.

In the first iteration, the planning module will first produce a subgoal to identify who the U.S. president in 2014 is. After passing the subgoal into the grounding module, we are able to acquire the corresponding query actions for execution. Once we get the executed results "*Obama*", the planning module would accept "*Obama*" along with the prior planning context as input to generate the next subgoal grounded in the current environment "*Subgoal 2: Query which state Obama was born*". Planning upon the latest execution results would mitigate the risk of introducing a non-existent object in the environment or a random entity during the reasoning process.

We demonstrate a single iteration of planning and grounding process of LUMOS-I in Figure 2b. In order to plan $t$-th subgoal, we input the 1) task description $T$, 2) prior subgoals $\{s_1, ..., s_{t-1}\}$, and 3) their executed results $\{e_1, ..., e_{t-1}\}$ into the planning module. We concatenate them in the format of $T, s_1, e_1, ..., s_{t-1}, e_{t-1}$ where the most recent subgoals and their results are placed in the end, as they have higher influence for planning $t$-th subgoal. The generated output would be the $t$-th subgoal, $s_t = \theta_{plan}(T, s_1, e_1, ..., s_{t-1}, e_{t-1})$. After the $t$-th subgoal is obtained, it will be directly incorporated into grounding module input together with the prior grounding history and action interface $I$ to generate the next set of executable actions $A_t$. Therefore, $A_t = \theta_{ground}(T, I, s_1, A_1, ..., s_{t-1}, A_{t-1}, s_t)$. Note that $A_t$ is an executable action list that includes one or multiple low-level actions, as the high-level subgoal can be decomposed into multiple low-level actions. We finally put the low-level actions $A_t$ into execution module and the final execution result $e_t$ can be sent back for planning $(t+1)$-th subgoal, $s_{t+1}$.

## 3 LEARNING TO PLAN & GROUND WITH OPEN-SOURCE LLMS

To guide planning and grounding modules to generate subgoals and valid low-level actions under our specified formulations, we fine-tune the two modules to produce the expected outputs.

Training the modules requires high-quality tasks, subgoals, and low-level actions. One solution to obtain these annotations is generating tasks, plans and grounded actions by LLMs from scratch. To equip smaller LLMs with instruction-following ability, prior works leverage methods such as Self-Instruct (Wang et al., 2023b) to synthesize the instructions, inputs and outputs based on seed examples. However, these methods are not suitable for generating high-quality annotations for complex interactive tasks. For example, GPT-4 can only achieve around 20% step success rate in a web agent benchmark Mind2Web (Deng et al., 2023; Liu et al., 2023a). Depending on such methods to generate complex interactive task annotations may introduce a great number of errors and degrade the annotation quality.

Instead of creating annotations with API-based LLMs directly, we exploit LLMs as a "style transfer" tool to convert ground-truth intermediate reasoning steps in existing benchmarks into the expected format in LUMOS formulations. We notice that there are several complex interactive tasks annotated with either human-written solutions or structured action sequences. For example, PRM800K (Lightman et al., 2023) is a maths dataset containing the solution steps written in natural language, interleaved with formulas; Musique (Trivedi et al., 2022) and StrategyQA (Geva et al., 2021) are complex QA datasets annotated with decomposed questions, supporting facts, and relevant Wikipedia paragraph indices; Mind2Web includes ground-truth action sequences such as "*[combobox] Reservation*

*Type → SELECT: Pickup*". They provide LLMs with ample fundamental information that sufficiently contributes to the annotation conversion.

## 3.1 Conversion Prompts

To help LLMs better follow the annotation conversion instructions, we add 4/5-shot in-context examples in conversion prompts (prompt details in Appendix F). We discuss the important properties of these in-context examples. The notations of all the converted annotations have hat over letters.

**Action Space.** Action space defines the available actions that LLMs could ground to. For example, for web agent annotations, we pre-define several common HTML operations in the action space: `Click`, `Type`, and `Select`.

**Ground-Truth Intermediate Reasoning Steps.** We provide LLMs with ground-truth intermediate reasoning steps in existing benchmarks. With these as a reference, LLMs are able to summarize high-level subgoals and synthesize corresponding actions according to the given action space.

**Subgoals and Corresponding Actions.** Subgoals and corresponding actions are denoted as $\hat{S}$ and $\hat{A}$ in the proposed formulations. When converting ground-truth reasoning steps into our expected annotations, it is necessary to provide LLMs with examples about how to distill the high-level subgoals from the reasoning steps and map them into corresponding actions. In the in-context examples, we manually decompose a complex task into several high-level subgoals according to the context of ground-truth reasoning steps. Under each high-level subgoal, we write down multiple corresponding actions that help to accomplish the subgoal (see examples in Appendix F). Given the aligned exemplar subgoals and actions in the prompt, LLMs would emulate to generate subgoals and their paired actions when converting annotations for new tasks.

As the executed results of prior subgoals might be useful in future action implementation, we interlink the grounded actions in the in-context examples to allow context-dependent execution. One typical example of the interlinked actions is R1 = KnowledgeQuery(Zombies); R2 = ParagraphRetrieve(R1, Query: What color skin are zombies typically depicted with?). The agent could first find the relevant paragraphs in the zombie knowledge page. Written in interlinked style, the second paragraph retrieval action is able to receive the knowledge about zombies (R1) as the context, and performs query-based retrieval.

**Intermediate Executed Results of Subgoals.** The intermediate executed results $\hat{E}$ play an important role in increasing LUMOS's adaptability to environmental changes. Some datasets (e.g., GSM8K, Mind2Web) offer execution results in their reasoning steps, including the computation results of formulas and the HTML code after operating on a website. We leverage them as $\hat{E}$.

For the datasets without any execution results, their reasoning steps actually contain the relevant documents that include the clues for solving subgoals. We take an annotated example in StrategyQA dataset. Although the direct answer of the decomposed question "*What color skin are zombies typically depicted with?*" is not provided, the annotation contains a related fact "*Zombies are often depicted as green in pallor.*" that mentions the answer "*green*". Thus, for each in-context example, we concatenate the relevant documents (e.g., "*Zombies are often depicted as green in pallor.*") as well as our manually captured executed results (e.g., "*green*") in the conversion prompts. When applying to converting new samples into our expected annotations, LLMs would automatically extract answers from the given documents as the executed results.

After prompting LLMs with the conversion prompts, we are able to acquire the key elements in training annotations, including subgoals $\hat{S}$, their corresponding actions $\hat{A}$ and execution results $\hat{E}$.

## 3.2 Transferring Annotations into Conversations

Finally, to build the interaction between planning and grounding modules, we organize the annotations into conversational format:

**Conversational Planning Module Annotation.** As shown in Figure 3a, we first play a user role to provide the task $\hat{T}$ and an instruction that guides the module to generate subgoals. For LUMOS-O

**Final planning module annotation**

```
Subgoal 1: Query the living period of Lowell Sherman.
 - Action 1-1: R1 = KnowledgeQuery(Lowell Sherman)
 - Action 1-2: R2 = ParagraphRetrieval(R1, Query: …)
 - Action 1-3: R3 = QA([R2], Query: …) = 46 years
Subgoal 2: Query the living period of Jonathan Kaplan.
 - Action 2-1: R4 = KnowledgeQuery(Jonathan Kaplan)
 - Action 2-2: R5 = ParagraphRetrieval(R4, Query: …)
 - Action 2-3: R6 = QA([R5], Query: …) = 75 years
Subgoal 3: Compare their living periods.
 - Action 3-1: R7 = Calculator(R3 > R6)
 - …
```

Organize subgoals & execution results into conversation format

```
<|user|>
Please provide a reasonable subgoal-based plan to solve the
given task.
Task: Who lives longer, Lowell Sherman or Jonathan Kaplan?

<|assistant|>
Subgoal 1: Query the living period of Lowell Sherman.

<|user|>
The execution result of Subgoal 1 is 46 years. Should we
keep planning?

<|assistant|>
No, I will keep planning. Subgoal 2: Query the living
period of Jonathan Kaplan.

<|user|>
The execution result of Subgoal 2 is 75 years. Should we
keep planning?
…
```

(a) Final planning module annotation organized from the converted subgoals & execution results.

**Final grounding module annotation**

```
Subgoal 1: Query the living period of Lowell Sherman.
 - Action 1-1: R1 = KnowledgeQuery(Lowell Sherman)
 - Action 1-2: R2 = ParagraphRetrieval(R1, Query: …)
 - Action 1-3: R3 = QA([R2], Query: …) = 46 years
Subgoal 2: Query the living period of Jonathan Kaplan.
 - Action 2-1: R4 = KnowledgeQuery(Jonathan Kaplan)
 - Action 2-2: R5 = ParagraphRetrieval(R4, Query: …)
 - Action 2-3: R6 = QA([R5], Query: …) = 75 years
Subgoal 3: Compare their living periods.
 - Action 3-1: R7 = Calculator(R3 > R6)
 - …
```

Organize subgoals & low-level actions into conversation format

```
<|user|>
Please ground the given subgoal to corresponding executable
actions for solving the given task.

[action space + action interfaces]

Task: Who lives longer, Lowell Sherman or Jonathan Kaplan?
Subgoal to be grounded: Subgoal 1: Query the living period
of Lowell Sherman.

<|assistant|>
R1 = KnowledgeQuery(Lowell Sherman); R2 =
ParagraphRetrieval(R1, Query: …); R3 = QA([R2], Query: …)

<|user|>
Subgoal to be grounded: Subgoal 2: Query the living period
of Jonathan Kaplan.

<|assistant|>
R4 = KnowledgeQuery(Jonathan Kaplan); …
…
```

(b) Final grounding module annotation organized from the converted subgoals & actions.

Figure 3: Process of converting converted subgoals, actions, and executions into the final conversational training annotations for LUMOS-I formulation.

formulation, the planning module should reply all the annotated subgoals $\hat{S}$ at once. There would be no further conversation needed.

LUMOS-I requires multi-turn conversational style. The planning module append the first ground-truth subgoal $\hat{s}_1$ with index "*Subgoal 1*" in the beginning. We then act as user again and put the executed results of $\hat{s}_1$ with prefix "*The executed result for Subgoal 1 is* ". The subsequent conversation is constructed in the similar patterns. We assume ourselves as user, tell the execution results $\hat{e}_{t-1}$ of the last subgoal $\hat{s}_{t-1}$ to planning module, and ask whether the planning should be stopped; The response would be whether the planning should stop and a new ground-truth subgoal $\hat{s}_t$.

**Conversational Grounding Module Annotation.** Shown in Figure 3b, we also first play a user role to provide the task $\hat{T}$, action space and interfaces $\hat{I}$. For LUMOS-O formulation, we feed all the subgoal annotations $\hat{S}$ in the first user prompt. All the action annotations $\hat{A}$ would be the response of the user instruction. For LUMOS-I formulation, we attach the first subgoal annotation $\hat{s}_1$ in the first user prompt. The response of grounding module should be the corresponding actions $\hat{A}_1$. In the rest conversations, we provide the current ground-truth subgoal $\hat{s}_t$, with prefix "*Subgoal to be grounded:* ". Its response would be $\hat{s}_t$'s corresponding actions $\hat{A}_t$.

### 3.3 TRAINING WITH CONVERTED ANNOTATIONS

As the style of LUMOS training annotations are conversational, we formulate them in the format of $\{x_1, y_1, ..., x_i, y_i, ..., x_n, y_n\}$, where $x_i$ and $y_i$ indicate $i$-th user prompts and their ground-truth responses. Following Wang et al. (2023a), during training process, we feed each entire multi-turn annotation into a decoder-only model while merely calculating the decoding loss on the tokens of ground-truth responses $Y = \{y_1, ..., y_i, ..., y_n\}$. We apply binary masking on the user prompt tokens to prevent computing loss on them. The final loss function $L = -\sum_j \log p_\theta(t_j \mid t_{<j}) \times \mathbf{1}(t_j \in Y)$ where $t_j$ denotes $j$-th input token and $\mathbf{1}(\cdot)$ is a Boolean indicator function.

# 4 EXPERIMENTS

We introduce the experimental setups, including the details of annotation conversion, training modules, and tools used in execution module. We then demonstrate the effectiveness of LUMOS via 1) comparing LUMOS with larger open-source LLM agents and GPT-4/3.5-based agent frameworks, 2) comparing with other potential open-sourced agent baselines, 3) evaluating LUMOS's generalizability on an unseen interactive task, and 4) assessing the quality of converted training annotations.

## 4.1 EXPERIMENTAL SETUPS

**Training Annotation Conversion.** We seek benchmarks of existing complex interactive tasks that comprise of ground-truth intermediate reasoning steps. Appendix A lists the data sources for annotation conversion. They encompass a variety of complex interactive tasks: maths, complex QA, and web agent tasks. Using the conversion prompts discussed in §3.1, we prompt GPT-4 OpenAI (2023) to perform annotation conversion on ground-truth reasoning steps. After filtering out invalid converted annotations with mismatched parentheses or too long lengths, in total, we obtain 39,441 and 39,558 annotations for training planning and grounding modules, respectively. Detailed statistics of the annotations in specific domains is shown in Appendix A.

**Planning and Grounding Module Training.** We use LLAMA-2-7B as the backbone model for planning and grounding module training. For all the experiments, we conduct training over two epochs using a learning rate of $2 \times 10^{-5}$. More training details are in the Appendix B.

**Action Spaces of Complex Interactive Tasks.** We incorporate the common actions frequently used for solving complex interactive tasks into the pre-defined action spaces. The action space of maths tasks includes `Calculator`, `SetEquation`, `SolveEquation`, `SolveInequality`, `Code`, `Define`, and `Count`; The action space of complex tasks incorporates `KnowledgeQuery`, `ParagraphRetrieval`, `QA`, `Calculator`, and `Code`; Web agent tasks involve the actions `Type`, `Click`, and `Select`. The execution tools for implementing each action are listed in Appendix D.

## 4.2 OVERALL PERFORMANCE ON COMPLEX INTERACTIVE TASKS

We evaluate LUMOS, open-source LLM agents and GPT-4/3.5-based agents on a variety of complex interactive tasks, including maths, complex QA and web agent tasks. We follow the evaluation settings adopted in AgentBench (Liu et al., 2023a) and ReWOO (Xu et al., 2023a) to assess LUMOS performance (see details in Appendix C). Table 1 shows the overall performance on each task type. Note that in Table 1, LUMOS-series agents whose names are free of subscripts (e.g., LUMOS-I) indicate that the agents are trained with the unification of all the annotations across domains. Agents like LUMOS-I$_{\text{webagent}}$ denote that the agents trained with the annotations exclusively derived from web agent tasks under LUMOS-I formulation.

**LUMOS vs. Larger Open-Source LLM Agents.** As LUMOS is based on open-source LLMs, we first compare LUMOS with other existing open-source LLM agents on the three complex interactive task types. We observe that LUMOS excels various open-source LLM agents over all the tested datasets. Although the base models of the compared language agents are approximately $2 - 4\times$ greater than LUMOS, LUMOS still outperforms them by a large margin. In particular, LUMOS-I achieves 24.6% success rate improvement over WizardLM-30B on Mind2Web.

**LUMOS vs. GPT-4/3.5-based Agents.** Though LUMOS is built upon small LLAMA-2-7B model, are they indeed much worse than strong GPT-4/3.5-based agents? Surprisingly, we discover that LUMOS is still able to perform better than those closed API-based agents. We find that LUMOS-I performs 5.1% superior to GPT-4 on Mind2Web while bringing 3.9% LLM accuracy improvement over GPT-3.5-based ReWOO agent on HotpotQA when using GPT-3.5-turbo as auxiliary QA tool.

## 4.3 COMPARISON WITH OTHER OPEN-SOURCE AGENT FORMULATIONS

To compare LUMOS with other open-source agents, we train open-source LLM agents in the following formulations with the same set of training annotations:

**Vanilla Training (Van-T)**: Given a task $T$, the agent learns to directly generate the answer.

(a) Performance on web agent tasks. The evaluation metric is step success rate (%).

| Agents | Web Agent |
|---|---|
| | Mind2Web |
| *Open-Source Language Agents* | |
| Baichuan-13B-chat[†] | 2.3 |
| WizardLM-30B[†] | 3.1 |
| Koala-13B[†] | 6.0 |
| *GPT-4/3.5-based Language Agents* | |
| GPT-3.5-turbo[†] | 15.7 |
| Claude[†] | 21.0 |
| GPT-4[†] | 22.6 |
| LUMOS-I$_{\text{webagent}}$ | 27.6 |
| LUMOS-I | **27.7** |

(b) Performance on maths tasks. The evaluation metric is accuracy (%).

| Agents | Maths | |
|---|---|---|
| | GSM8K | SVAMP |
| *Open-Source Language Agents* | | |
| Code-Llama (PoT)-13B[¶] | 36.1 | 60.0 |
| Platypus-30B[¶] | 37.8 | 51.7 |
| ReWOO-Planner-7B[‡] | ∼38 | - |
| Orca-Platypus-13B[¶] | 38.4 | 56.9 |
| Alpaca-7B[‡] | ∼39 | - |
| Galactica-30B[¶] | 41.7 | 41.6 |
| LUMOS-O$_{\text{maths}}$ | **50.5** | **65.5** |
| LUMOS-I$_{\text{maths}}$ | 47.1 | 63.6 |
| LUMOS-I | 46.7 | 63.8 |

(c) Performance on complex QA tasks. The evaluation metric for StrategyQA and HotpotQA is accuarcy (%) and LLM accuracy (%), respectively.

| Agents | Agent Base Model | QA Tool | Complex QA | |
|---|---|---|---|---|
| | | | StrategyQA | HotpotQA |
| *Open-Source Language Agents* | | | | |
| ReWOO-LLAMA[‡] | LLAMA-7B | GPT-3.5-turbo | ∼56 | ∼37 |
| *GPT-4/3.5-based Language Agents* | | | | |
| GPT-3.5-CoT[‡] | GPT-3.5-turbo | GPT-3.5-turbo | 56.0 | 37.8 |
| ReAcT[‡] | GPT-3.5-turbo | GPT-3.5-turbo | 64.6 | 40.8 |
| ART* | GPT-3 | GPT-3 | 66.4 | - |
| ReWOO[‡] | GPT-3.5-turbo | GPT-3.5-turbo | 66.6 | 42.4 |
| LUMOS-O$_{\text{complexQA}}$ | LLAMA-2-7B | GPT-3.5-turbo | 60.6 | 39.2 |
| LUMOS-I$_{\text{complexQA}}$ | LLAMA-2-7B | GPT-3.5-turbo | 65.7 | 45.9 |
| LUMOS-I$_{\text{complexQA}}$ | LLAMA-2-7B | GPT-4 | **72.4** | **56.8** |
| LUMOS-I | LLAMA-2-7B | GPT-3.5-turbo | 66.3 | 46.3 |

(d) Performance on unseen task, WebShop. The evaluation metric for WebShop is the average reward defined in Yao et al. (2022a).

| Agents | Unseen Task |
|---|---|
| | WebShop |
| *Open-Source Language Agents* | |
| Baichuan-13B-chat[†] | 5.7 |
| Koala-13B[†] | 6.0 |
| WizardLM-30B[†] | 10.6 |
| Vicuna-13B[†] | 12.6 |
| *API-based Language Agents* | |
| ChatGLM2[†] | 19.4 |
| LUMOS-I$_{\text{webagent}}$ | 34.7 |
| LUMOS-I$_{\text{maths}}$ | 30.1 |
| LUMOS-I$_{\text{complexQA}}$ | 33.5 |
| LUMOS-I | **39.8** |

Table 1: Overall performance on diverse complex interactive tasks. [†], [‡], [*], and [¶] denote the results reported in Liu et al. (2023a), Xu et al. (2023a), Paranjape et al. (2023) and Yue et al. (2023). The symbol ∼ means that the performance is estimated according to the bar chart in ReWOO's Figure 6.

**Chain-of-Thought Training (CoT-T)**: Given a task $T$, the agent learns to directly produce the chain-of-thought solution as well as the final answer.

**Unmodularized Agent Training (UA-T)**: Given a task $T$, the agent learns to generate all the sub-goals and corresponding actions with a single LLM-based module. Execution module is served for executing the grounded actions.

Note that we do not compare LUMOS formulations with vanilla and CoT training on web agent tasks, as without any interaction with external environment, their planning is not grounded to the real websites. Also, because Van-T and CoT-T cannot be successfully applied to web agent task, we choose not to train the models with unified annotations that cover web agent task annotations; instead, all the models would be trained with each specific task type's annotations.

(a) Comparison with other training formulations on web agent tasks.

| Agents | Web Agent |
|---|---|
| | Mind2Web |
| UA-T | 25.3 |
| LUMOS-I$_{\text{webagent}}$ | **27.6** |

(b) Comparison with other training formulations on maths tasks.

| Agents | Maths | |
|---|---|---|
| | GSM8K | SVAMP |
| Van-T | 9.9 | 12.7 |
| CoT-T | 40.4 | 52.2 |
| UA-T | 45.5 | 61.7 |
| LUMOS-O$_{\text{maths}}$ | **50.5** | **65.5** |
| LUMOS-I$_{\text{maths}}$ | 47.1 | 63.6 |

(c) Comparison with other training formulations on complex QA tasks.

| Agents | Complex QA | |
|---|---|---|
| | StrategyQA | HotpotQA |
| Van-T | 61.0 | 23.4 |
| CoT-T | 58.3 | 22.1 |
| UA-T | 62.3 | 39.6 |
| LUMOS-O$_{\text{complexQA}}$ | 60.6 | 39.2 |
| LUMOS-I$_{\text{complexQA}}$ | **65.7** | **45.9** |

Table 2: Comparison with baseline formulations for training language agents.

**LUMOS vs. Van-T and CoT-T.** From Table 2, we find that LUMOS-I and LUMOS-O both outperform the formulations Van-T and CoT-T by a large margin. Building the connections with execution modules, LUMOS can receive more reliable intermediate results.

**LUMOS vs. UA-T.** We notice that the two LUMOS formulations also perform better than the unmodularized formulation that makes use of only one model. It further shows the benefits of disentangling different agent skills and simplifying the learning tasks for language agents.

### 4.4 GENERALIZABILITY TO DIVERSE COMPLEX INTERACTIVE TASK

To evaluate the generalizability of LUMOS, we focus on answering the two questions — **Q1: Could unified training further improve the training task types? Q2: Could unified training achieve better performance on an unseen task than training with task-specific annotations?** Here, we select WebShop as the unseen task, since its shopping environment and action space greatly differs from the ones covered in the training annotations. To adapt LUMOS to the new task, we add two-shot examples in the input of planning and grounding modules for them to learn how to produce subgoals and ground to new sets of available actions (see more details in Appendix G).

From Table 1, it is shown that LUMOS-I, the product of unified training, slightly enhances the domain-specific agents on web agent and complex QA tasks. More importantly, as displayed in Table 1d, LUMOS-I achieves 5-10 average reward than the domain-specific agents on WebShop. Meanwhile, it significantly excels larger language agents such as WizardLM-30B and Vicuna-13B (Chiang et al., 2023). This manifests that unified training would not hinder the performance on the trained tasks, while equipping agents with better capacity to plan for new tasks and understand novel actions.

In the end, we analyze the quality of our converted training annotations in Appendix E. We show that 1) our proposed annotation conversion method can generate annotations that bring better performance than using Self-Instruct method, and 2) the design of having high-level subgoal annotations is beneficial than using low-level subgoal to supervise the agent training.

## 5 RELATED WORK

**Language Agents.** Empowered by reinforcement learning, language agents were previously deployed in text game environments such as TextWorld (Côté et al., 2019) and LIGHT (Urbanek et al., 2019). With the development of LLMs, language agents demonstrate the potential to solve more diverse complex interactive tasks. ReAct (Yao et al., 2022b) is a prompting method that makes LLM-based language agents grounded in external environment and aware of the generated action feedback for further reasoning. Subsequently, various methods (e.g., HuggingGPT (Shen et al., 2023), Chamelon (Lu et al., 2023), ReWOO (Xu et al., 2023a), ART (Paranjape et al., 2023), and BOLAA (Liu et al., 2023b)) aimed to improve agent performance and make agents applicable to more diverse scenarios. However, these recent language agent works mainly rely on prompting large closed LLMs such as OpenAI GPT-4/3.5. Different from the prior work, we explore whether our training paradigm enables smaller open-source LLMs to achieve comparable performance.

**Improving Capacity of Smaller Models.** Knowledge distillation (Hinton et al., 2015) is a prevalent technique to transfer knowledge from teacher models to smaller models for better efficiency. Recent works utilize LLMs to generate explicit textual knowledge or training data for training smaller models (Bosselut et al., 2019; West et al., 2022; Wang et al., 2023b; Shridhar et al., 2023; Li et al., 2023). We notice that directly generating annotations with for training planning and grounding modules may introduce large number of errors because several strong LLMs (e.g., GPT-4) achieve very low performance on some of the complex interactive tasks (Liu et al., 2023a). In our formulation, instead of using LLMs to create training annotations without any reference, LLMs transform the gold reasoning steps into our desired annotation format for training system modules.

## 6 CONCLUSION

We introduce LUMOS, **L**anguage agents with **U**nified formats, **M**odular design, and **O**pen **S**ource LLMs. We propose two training formulations LUMOS-I and LUMOS-O that encourage collaboration between planning, grounding, and execution modules to solve complex tasks. To obtain high-quality annotations for training modules, we leverage LLMs to convert reasoning steps in existing benchmarks into a unified format usable in the LUMOS framework. Merely built upon small LLMs, LLAMA-2-7B, LUMOS is competitive with the LLM agents that are $2 - 4\times$ larger on maths tasks, and even outperforms GPT-4/3.5-based agents on complex QA and web agent tasks. We also show that LUMOS achieves superior performance against commonly adopted open-source agent baseline formulations and enjoys considerably better generalization on an unseen interactive task.

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

APPENDIX

## A    STATISTICS OF CONVERTED TRAINING ANNOTATIONS

As discussed in §4.1, the data sources for constructing training annotations cover a broad range of complex interactive tasks. Table 3 shows the benchmarks leveraged for annotation conversion, along with the task type information.

To train agents like LUMOS-I$_{\text{maths}}$ mentioned in Table 1b, we need to leverage the annotations converted from 19778 data specific to maths domain. For training a unified agent such as LUMOS-I, we would use the annotations transformed from all the listed data as training set.

## B    DETAILS OF TRAINING MODULES

We describe additional details about our training experiments. We set the maximum sequence length to 1024. We also apply linear warmup for 3% of the total training steps to adjust the learning rate. All the training experiments are implemented with 2 NVIDIA 80GB A100 GPUs.

| Task Types | Datasets | Numbers for Conversion | Total Numbers |
|---|---|---|---|
| Maths | PRM800K | 10000 | |
| | GSM8K | 7473 | 19778 |
| | ASDiv | 2305 | |
| Complex QA | Musique | 17632 | 19409 |
| | StrategyQA | 1777 | |
| Web Agent | Mind2Web | 1009 | 1009 |

Table 3: Statistics of data sources used for converting annotations.

## C    DETAILS OF PERFORMANCE EVALUATION

**Metrics.** Here we mainly discuss the special metrics adopted to evaluate the agent performance. For HotpotQA, instead of using strict exact matching, we follow Xu et al. (2023a) to use GPT-4 as an evaluator to judge whether the predicted answer shares the same semantics with the gold answer. We call this metric as LLM accuracy, frequently mentioned in §4. For Mind2Web, we adopt the same metric step success rate used for AgentBench evaluation. A step is deemed successful solely when both the chosen HTML tag and predicted action type exactly match the gold action. For WebShop, we leverage the reward utilized in both AgentBench and original WebShop paper, which quantify the similarity between gold and predicted products with regard to product titles and selected attributes.

**Evaluation Data.** Following Xu et al. (2023a), we only evaluate 300 and 1000 randomly selected examples from StrategyQA and HotpotQA evaluation set, respectively. The results reported in Table 1c are the average performance on three different sets of sampled data. Regarding Mind2Web, we only evaluate on the "cross-domain" test set that AgentBench utilizes for evaluation. For WebShop, we evaluate the first 500 instances from the entire test set as AgentBench used to do.

## D    EXECUTION TOOLS ASSOCIATED WITH ACTION SPACES

For each available action defined in the action spaces, there are at least one associated backend execution tools that help to implement the actual grounded actions. For maths tasks, the main execution tool is WolframAlpha API [2] as it is capable of performing a large collection of mathematical functions such as calculating formulas and solving equations. For complex mathematical operations such as sorting, we would leverage OpenAI Codex (Chen et al., 2021) to generate a short code snippet for execution. For complex QA tasks, we rely on Wikipedia and Google search to help us locate relevant entity knowledge. Besides, we leverage a pre-trained semantic matching model

---

[2]https://www.wolframalpha.com/.

`dpr-reader-multiset-base`[3] used in Dense Passage Retrieval (DPR) (Karpukhin et al., 2020) to capture relevant paragraphs according to the given query. Following ReWOO (Xu et al., 2023a), we also include GPT-series model as a simple QA tool to answer the query based on our retrieved knowledge. For web agent tasks, the actions are real mouse and keyboard operations including typing, clicking and selecting HTML tags. To locate the relevant HTML tags to be operated, following AgentBench evaluation, we use a pre-trained DeBERTa model[4] that ranks and retrieves the tags according to the current action we would perform.

## E  FURTHER ANALYSIS ON TRAINING ANNOTATIONS

In the annotation analysis part, we aim to answer two questions regarding the quality and format decision. **Q1: How good is our converted training annotations? Q2: Would it be better if we adopt low-level subgoals instead of our proposed high-level subgoals?**

**Annotation Quality Assessment.** We evaluate the annotation quality by checking the agent performance after training models with the annotations. Table 4 has shown the competitiveness of LUMOS, implying the high quality of our annotation data. Furthermore, we compare with ReWOO-Planner annotations, another training annotations to train language agents, constructed upon HotpotQA and TriviaQA datasets with Self-Instruct method. They discard all the annotations that cannot lead to the ground-truth answers to guarantee the annotation quality. For fair comparison, we train ReWOO-Planner's base model, LLAMA-7B

| Training Data | Complex QA | |
|---|---|---|
| | StrategyQA | HotpotQA |
| *Downstream Perf. of Training Different Data* | | |
| ReWOO-Planner Data | ∼57 | ∼37 |
| LUMOS-I$_{complexQA}$ Data | **58.3** | **38.1** |
| *Perf. Using High-Level and Low-Level Subgoal Annots.* | | |
| LUMOS-I$_{complexQA}$ w/ Low-Level Subgoals | 63.3 | 44.3 |
| LUMOS-I$_{complexQA}$ Data | **65.7** | **45.9** |

Table 4: Comparison between the agents trained with different annotations.

with our training annotations. As the size of ReWOO-Planner annotations is 2,000, we also sample 2,000 data from our obtained annotations for fair comparison. Since ReWOO-Planner data is fully based on QA benchmarks, we focus on the comparison on complex QA tasks.

Displayed in Table 4, we find that our training annotations can bring approximately 2.3% accuracy and 1.1% LLM accuracy improvement over ReWOO-Planner on StrategyQA and HotpotQA, respectively. In particular, while ReWOO-Planner data is built upon HotpotQA, it still cannot help to achieve better performance than our annotations on the in-domain HotpotQA data. It further suggests that our proposed annotation conversion method is effective.

**Low-Level Subgoal vs. High-Level Subgoal.** As discussed in §2, we encourage LLMs to generate high-level subgoals that correspond to one or multiple low-level actions. A natural alternative annotation could be that each subgoal corresponds to only one low-level action, i.e., the subgoal can also be treated as "low-level". We prompt LLMs to generate the annotations with low-level subgoals by changing the in-context examples to the format where a subgoal can only ground to exactly one action. We perform analysis on complex QA tasks, since each subgoal in converted QA annotations corresponds to more than two low-level actions. As shown in Table 4, we find that the performance drops after switching high-level subgoals into low-level ones on both QA datasets. This further validates the choice of subgoal design.

## F  IN-CONTEXT EXAMPLES IN CONVERSION PROMPTS

As discussed in §3.1, in-context examples are helpful to instruct LLMs to generate annotations in our expected format. For each of the training task types, we showcase one in-context example to help readers better understand how the prompting conversion method works and the format of our expected annotations. We highlight subgoals and their corresponding actions and execution results with yellow, red and blue, respectively.

---

[3]https://huggingface.co/facebook/dpr-reader-multiset-base.

[4]https://huggingface.co/osunlp/MindAct_CandidateGeneration_deberta-v3-base.

## F.1    IN-CONTEXT EXAMPLE FOR OBTAINING MATHS TASK ANNOTATIONS

Please convert natural language plans into a series of subgoals and their corresponding actions that lead to the successful implementation with respect to the given instructions. Please use 'R[number]' to represent the intermediate results for each subgoal, without generating any exact values. Please also use functions to represent the corresponding actions. For the actions, they must be one of 'Calculator', 'SetEquation', 'SolveEquation', 'SolveInequality', 'Count', 'Code', and 'Define'.

Example 1:

Task: Peter goes to the store to buy a soda. The soda costs $.25 an ounch. He brought $2 with him and leaves with $.50. How many ounces of soda did he buy?

Natural language plan:
He spend $1.5 on soda because 2 - .5 = 1.5 He bought 6 ounces of soda because 1.5 / .25 = 6

Subgoal-based plan:
Subgoal 1: Calculate how much the soda costs in total.
Action 1-1: R1 = Calculator(2 - 0.5) = 1.5

Subgoal 2: Calculate the ounces of soda the price per ounch.
Action 2-1: R2 = Calculator(R1 / 0.25) = 6

## F.2 IN-CONTEXT EXAMPLE FOR OBTAINING COMPLEX QA TASK ANNOTATIONS

Please convert natural language plans into a series of subgoals and their corresponding actions that lead to the successful implementation with respect to the given instructions. Please use 'R[number]' to represent the intermediate results for each subgoal, without generating any exact values. Please also use functions to represent the corresponding actions. For the actions, they must be one of one of 'KnowledgeQuery', 'ParagraphRetrieve', 'QA', 'Calculator' and 'Code'.

Example 1:

Task: Are more people today related to Genghis Khan than Julius Caesar?

Natural language plan:
We find relevant facts: Julius Caesar had three children. Genghis Khan had sixteen children. Modern geneticists have determined that out of every 200 men today has DNA that can be traced to Genghis Khan. We need to answer these questions: 1. How many kids did Julius Caesar have? (Can be answered based on paragraph 'Julius Caesar-75') 2. How many kids did Genghis Khan have? (Can be answered based on paragraph 'Genghis Khan-17') 3. Is #2 greater than #1? Based on these evidences and decomposed questions, the answer is True.

Subgoal-based plan:
Subgoal 1: Obtain the number of the kids that Julius Caesar had.
Action 1-1: R1 = KnowledgeQuery(Julius Caesar) = WikipediaPage(Julius Caesar)
Action 1-2: R2 = ParagraphRetrieve(R1, Query: How many kids did Julius Caesar have?) = Paragraph(Julius Caesar-75).
Action 1-3: R3 = QA([R2], Question: How many kids did Julius Caesar have?) = 3.

Subgoal 2: Obtain the number of the kids that Genghis Khan had.
Action 2-1: R4 = KnowledgeQuery(Genghis Khan) = WikipediaPage(Genghis Khan).
Action 2-2: R5 = ParagraphRetrieve(R4, Query: How many kids did Genghis Khan have?) = Paragraph(Genghis Khan-17).
Action 2-3: R6 = QA([R5], Question: How many kids did Genghis Khan have?) = 16.

Subgoal 3: Determine if Genghis Khan had more kids.
Action 3-1: R7 = Calculator(R6 > R3) = True

## F.3  IN-CONTEXT EXAMPLE FOR OBTAINING WEB AGENT TASK ANNOTATIONS

Since the data source for converting annotations, Mind2Web, already provides the ground-truth execution results after each action, as discussed in §3.1, we do not ask LLMs to capture each action's execution results. Therefore, there are no parts highlighted with blue in the in-context example.

```
Please convert natural language plans into a series of subgoals and their
corresponding actions that lead to the successful implementation with respect
to the given instructions. Please use 'R[number]' to represent the intermediate
results for each subgoal, without generating any exact values. Please also use
functions to represent the corresponding actions. For the actions, they must be
one of they must be one of 'TYPE', 'CLICK', and 'SELECT'.

Example 1:

Task: Find a Ricky Kej track to listen and share which has been added in the last
year and is between 2 to 10 minutes.

Natural language plan:
[searchbox] Search ⟶ TYPE: Ricky Kej; [link] Search for ''Ricky Kej'' ⟶
CLICK; [link] Tracks ⟶ CLICK; [link] Added any time ⟶ CLICK; [link] Past year
⟶ SELECT; [link] Any length ⟶ CLICK; [link] 2-10 min ⟶ CLICK; [link] To
listen to ⟶ CLICK; [link] To share ⟶ CLICK

Subgoal-based plan:
Subgoal 1: Type Ricky Kej to search his songs.
Action 1-1: R1 = TYPE(Env, QUERY: Type Ricky Kej to search his songs, TEXT: Ricky
Kej)

Subgoal 2: Click on the option to search for Ricky Rej.
Action 2-1: R2 = CLICK(R1, QUERY: Click on the option to search for Ricky Rej)

Subgoal 3: Choose tracks as the search category.
Action 3-1: R3 = CLICK(R2, QUERY: Choose tracks as the search category)

Subgoal 4: Find the region to adjust the added time of our interested track.
Action 4-1: R4 = CLICK(R3, QUERY: Find the region to adjust the added time of our
interested track)

Subgoal 5: Choose the last year as the added date.
Action 5-1: R5 = SELECT(R4, QUERY: Choose the last year as the added date, TEXT:
Past year)

Subgoal 6: Find the region to adjust the track length of our interested track.
Action 6-1: R6 = CLICK(R5, QUERY: Find the region to adjust the track length of
our interested track)

Subgoal 7: Choose 2 to 10 minutes as the track length.
Action 7-1: R7 = CLICK(R6, QUERY: Choose 2 to 10 minutes as the track length)

Subgoal 8: Listen to our searched track.
Action 8-1: R8 = CLICK(R7, QUERY: Listen to our searched track)

Subgoal 9: Share our searched track.
Action 9-1: R9 = CLICK(R8, QUERY: Share our searched track)
```

# G  IN-CONTEXT EXAMPLE FOR ADAPTING MODULES TO UNSEEN TASK — WEBSHOP

As discussed in §4.4, the in-context examples are used for helping modules understand how to plan and ground to novel available actions. We showcase one in-context WebShop-related example used for the inference of planning and grounding modules, respectively.

## G.1  IN-CONTEXT WEBSHOP-RELATED EXAMPLE FOR PLANNING MODULE

```
<|user|>
Please provide a reasonable subgoal-based plan to solve the given task.
Task: Find me slim fit, loose fit women's tops, tees & blouses with long sleeve,
short sleeve with color: a01#black, and size: large, and price lower than 50.00
dollars

<|assistant|>
Subgoal 1: Search two relevant items and the product names according to the
instruction.

<|user|>
The execution result for Subgoal 1 is b09s3bn15c - Mens Linen Shirt,Men's Striped
Shirts Casual Short Sleeve Button Down Shirts Regular Fit Hawaiian Shirts Beach
Tees Tops ** b094q7b3ss - Women Cold Shoulder Tops, Summer Butterfly Print Shirts
Fashion Casual Short Sleeve Plus-Size Tunic Top Tee and Blouse.

<|assistant|>
Subgoal 2: Select the most relevant features of item b09s3bn15c.

<|user|>
The execution result for Subgoal 2 is short, casual, shoulder.

<|assistant|>
Subgoal 3: Select the most relevant features of item b094q7b3ss.

<|user|>
The execution result for Subgoal 3 is black, large, x-large.

<|assistant|>
Subgoal 4: Pick up the most related one from the two relevant items according to
the product names and their features.

<|user|>
The execution result for Subgoal 4 is b094q7b3ss.

<|assistant|>
Subgoal 5: Buy this item by selecting its relevant features.

<|user|>
The execution result for Subgoal 5 is Done.
```

## G.2 IN-CONTEXT WEBSHOP-RELATED EXAMPLE FOR GROUNDING MODULE

```
<|user|>
Please ground the given subgoal to corresponding executable actions for solving
the given task. The grounded actions must be the one in available action list.

The available action list is 'Search', 'FeatureRetrieval', 'Pick' and
'Click'.   Search(Query):  Search related items according to the Query;
FeatureRetrieval(Feature_list, Query): Select the most relevant features from
Feature_list according to Query; Pick(Item_names, Item_features, Query): Pick
the most relevant item from Item_names according to Query, Item_names and
Item_features; Click(Item): Click the Item to check more information.

Task: Find me slim fit, loose fit women's tops, tees & blouses with long sleeve,
short sleeve with color: a01#black, and size: large, and price lower than 50.00
dollars
Subgoal to be grounded: Subgoal 1: Search two relevant items and the product
names according to the instruction.

<|assistant|>
R1 = Search(Find me slim fit, loose fit women's tops, tees & blouses with long
sleeve, short sleeve with color: a01#black, and size: large, and price lower than
50.00 dollars)

<|user|>
Subgoal to be grounded: Subgoal 2: Select the most relevant features of item
b09s3bn15c.

<|assistant|>
R2 = Click(b09s3bn15c); R3 = FeatureRetrieval(R2, Query: Find me slim fit, loose
fit women's tops, tees & blouses with long sleeve, short sleeve with color:
a01#black, and size: large, and price lower than 50.00 dollars)

<|user|>
Subgoal to be grounded: Subgoal 3: Select the most relevant features of item
b094q7b3ss.

<|assistant|>
R4 = Click(< prev); R5 = Click(b094q7b3ss); R6 = FeatureRetrieval(R5, Query: Find
me slim fit, loose fit women's tops, tees & blouses with long sleeve, short sleeve
with color: a01#black, and size: large, and price lower than 50.00 dollars)

<|user|>
Subgoal to be grounded: Subgoal 4: Pick up the most related one from the two
relevant items according to the product names and their features.

<|assistant|>
R7 = Pick(Item_names: R1, Item_features: [R3, R6], Query: Find me slim fit,
loose fit women's tops, tees & blouses with long sleeve, short sleeve with color:
a01#black, and size: large, and price lower than 50.00 dollars)

<|user|>
Subgoal to be grounded: Subgoal 5: Buy this item by selecting its relevant
features.

<|assistant|>
R8 = Click(< prev); R9 = Click(R7); R10 = Click(black); R11 = Click(large); R12
= Click(x-large); R13 = Click(buy now)
```

