# OpenReview forum: "LUMOS: Towards Language Agents that are Unified, Modular, and Open Source"
_ICLR.cc/2024/Conference — Submitted to ICLR 2024_

### Official Review · Reviewer_xkWX · 2023-11-01

**Soundness:** 3 good
**Presentation:** 3 good
**Contribution:** 3 good
**Rating:** 6
**Confidence:** 3

**Summary:**

This paper presents an overview of the LLM Agent architecture, with: planning, execution and grounding. The method is competitive for math, QA, and WebShop tasks with significantly smaller model size. The proposed framework is generally applicable to tasks where Language Models are used as an agent.

**Strengths:**

1. The paper demonstrated a solid framework for using smaller models as LLM agents.
2. The presentation of the paper is easy to follow and the figure is straightforward.
3. I do appreciate the paper does some detailed ablations of the method, making it stronger.

**Weaknesses:**

1. My first question for the authors is: the agent framework has been discussed quite often (there have been some follow-ups since the ReACT paper came out). Although the author claimed they are mostly based on close-source API based models (not the open-source ones), it seems the architecture is quite similar?
2. Why on math tasks, LUMOS-O significantly outperforms LUMOS-I while in the other two tasks, the results seem to reverse? Some more analysis on the error patterns would be preferred to give some more insights.
3. Why LUMOS in general outperforms UA-T? Does this mean some tasks jointly fine-tuned together can result in some conflict? Does this imply that we should train multiple smaller models each fine-tuned for a specific task?

**Questions:**

Please see above for comments.

---

> ### Author Response · Authors · 2023-11-20
> **Response to Reviewer xkWX**
>
> Dear Reviewer xkWX,
>
> Thank you for your thoughtful review. We appreciate your note that we contribute a solid framework to help even smaller models achieve success as language agents. Below we answer your questions:
>
> ---
>
> ### **1. Similar architecture to existing agent framework**
>
> Rather than merely presenting a plan-ground-execute modular architecture, our research is more inclined towards exploring two key areas: 1) developing suitable formulations to effectively train a superior open-source language agent, and 2) studying how to construct training annotations that are in line with our proposed formulations for planning and grounding modules.
>
> Regarding the training formulations, very few previous studies on agent fine-tuning studies which formulations might enhance agent training. Many established agent frameworks (like ReAct and ReWOO) depend solely on a single model for both planning and grounding tasks. Whereas, we are unsure about the real efficacy of using a single model for learning both planning and grounding skills.
>
> Moreover, there's a possibility that simple chain-of-thought fine-tuning might suffice for complex tasks such as math and complex QA tasks [1]. Open-source LLMs such as LLAMA-2 have already learned a substantial amount of basic math and general knowledge during their pre-training phases. Through chain-of-thought fine-tuning, these models could potentially acquire planning skills fast, and utilize their stored knowledge without any external math and QA tools. As indicated in Section 4.3, we uncover the performance of various open-source agent training formulations. This could serve as **a practical guide for developing more advanced open-source agents** in the future.
>
> [1] Distilling Reasoning Capabilities into Smaller Language Models. Shridhar et al., 2023. ACL Findings 2023.
>
> ---
>
> ### **2. Analysis on the error patterns of Lumos-O and Lumos-I on math tasks**
>
> By observing the cases where Lumos-I makes mistakes, we find that Lumos-I is slightly worse than Lumos-O to terminate the planning. A typical example is shown as follows. There’s a math problem: “James decides to run 3 sprints 3 times a week.  He runs 60 meters each sprint.  How many total meters does he run a week?” Lumos-I will only solve part of the problem by merely calculating 3*60=180, while Lumos-O could generate the perfect solution. Another example is “Kylar went to the store to buy glasses for his new apartment. One glass costs $5, but every second glass costs only 60% of the price. Kylar wants to buy 16 glasses. How much does he need to pay for them?” Lumos-I will repeatedly generate subgoals “Calculate the cost of the third/fourth/fifth/sixth … glass.”
>
> Another possible reason why Lumos-I is not better than Lumos-O lies in the ineffectiveness of the intermediate reasoning subgoals formulated by Lumos-I for the generation of subsequent subgoals. Consider the aforementioned mathematical problem such as, “James decides to run 3 sprints 3 times a week. He runs 60 meters each sprint. How many total meters does he run a week?” Even if the agent calculated how many times James sprints a week, which is 3*3=9, the mere number 9 does not affect the next subgoal generation, which could be “Calculate the total meters James runs a week”, without any mentioning of the previous calculation results.
>
> ---
>
> ### **3. Discussion about the comparison between Lumos and UA-T**
>
> UA-T represents a training approach that teaches a single LM to simultaneously handle both planning and grounding tasks. However, due to the substantial differences between the goals of planning and grounding, training a single model on both skills might impede the development of each skill individually.
>
> Moreover, incorporating grounding instructions and processes into the planning stages significantly lengthens the model's input data. When generating the last subgoal, the input size for UA-T is 359.5 words, 2.63 times longer than the original input dimensions used for the Lumos planning module. When generating a new subgoal, this increase in input length could lead to the model partially losing track of earlier planning contexts. Consequently, this might adversely affect the accuracy of subsequent subgoal generation.
>
> ---
>
> ### **Thank you!**
>
> We're grateful for your thoughtful questions and helpful advice. Please feel free to reach out if you have any additional queries. If you feel that our response adequately addresses your concerns, would you like to consider raising your rating score for our paper? Thank you for your consideration!

---

> > ### Author Response · Authors · 2023-11-21
> > **Gentle Reminder**
> >
> > Dear Reviewer,
> >
> > We hope you are doing well. As the discussion period concludes tomorrow (Nov. 22nd), we wanted to reach out to see if you have any follow-up questions. If so, we would appreciate the opportunity to respond before the discussion period ends. We believe our above messages should have addressed your concerns, and therefore may warrant an increase in score if you find them helpful as well. Would you please let us know if you have any further questions or concerns? We are happy to continue the discussion.
> >
> > Thank you very much again for your thoughtful review and help in improving the paper. We appreciate your time and consideration.
> >
> > Best regards,
> >
> > Authors of Lumos

---

> > ### Author Response · Authors · 2023-11-23
> > **Kindly Reminder (2)**
> >
> > Dear Reviewer xkWX,
> >
> > We hope you are doing well. With the incoming discussion deadline, we wanted to check if you have any additional inquiries. If there are any follow-up questions on your end, we would welcome the opportunity to address them before the conclusion of the discussion period. If our previous correspondences have adequately resolved your concerns, we kindly inquire if you would consider raising your score. We appreciate your time and consideration.
> >
> > During the last day, one reviewer raised 2 scores, and another reviewer acknowledged we have addressed his/her concerns. Furthermore, in our latest general response, we have presented additional evidence highlighting Lumos' enhanced generalizability on a novel task, InterCode-SQL. Please don't hesitate to reach out if you have any further inquiries.
> >
> > Best regards,
> >
> > Authors of Lumos

---

### Official Review · Reviewer_7VK9 · 2023-11-01

**Soundness:** 2 fair
**Presentation:** 2 fair
**Contribution:** 2 fair
**Rating:** 5
**Confidence:** 2

**Summary:**

The paper proposes a new framework to train LLMs for certain tasks such as answering questions related to maths, textual comprehension and outputting actions for a website (click). The new framework has three parts (“modules”):

- “planning”, which converts a prompt to (simpler, but still human language-like) queries (“subgoals”), e.g. “Query the living period of Lowell Sherman”

- “grounding”, which converts the subgoals to “actions”: function-call type queries, e.g. “KnowledgeQuery(Lowell Sherman)”

- “execution”, which executes those actions

For both planning and the grounding phases (“modules”) GPT-4 is used to generate annotated examples for the given task at hand to train LLAMA-7B with.

Two versions are suggested: LUMOS-O (which goes through the above three parts/modules sequentially) and LUMOS-I (which iterates through each subgoal until the execution of that subgoal, the result of which is then used for the next planning the next subgoal).The framework is applied to the open source LLAMA 7B LLM and claims superior or competitive performance on larger LLMs without or with techniques to improve their performance such as Chain of Thought prompting, SelfInstruct, ReWOO-Planner-7B (an improvement of React) and for this claim provides experiment details on these tasks.

**Strengths:**

The paper proposes a new framework/pipeline to achieve better results with prompting LLMs in a nascent field, which could be seen as original.

Quality: The methodology of the paper is well-documented, the experiment section contains results on relevant datasets.

Clarity: the paper follows the ICLR formatting style, images are mostly clearly captioned, there is an attempt to place the work in the (very recent) literature. Large sections of the paper are easy to understand.

In terms of significance, using their particular setting a 7B-parameter open-source LLM outperforms larger LLMs queried by previous techniques.

**Weaknesses:**

Having said the above in terms of strengths, in my opinion the paper includes a fair bit of weaknesses.

The paper’s claim that LUMOS outperforms or is competitive compared to larger LLMs is not well justified by evidence. A list of issues are:

- F1-scores are not reported, and to the best of my knowledge that is the primary metric in this field of AI

- The LLM is fine-tuned for the specific tasks (for subgoal and action generation), whereas the baselines use publicly available APIs at best, hence they are more general

- Unless I am mistaken: on GSM8K, ReWOO achieves 62.4% accuracy (as opposed to the reported ~38) in Xu et al. 2023 and is significantly better than LUMOS (50.5%)

- I could be wrong, but to me it looks like ReWOO trained a 7B model, they only used GPT-3.5 for the QA tool. I also cannot find the claimed LLAMA-7B results in Xu et al. 2023. Also, ReWOO-7B uses GPT-3.5-turbo as a QA tool, achieves 66.6% accuracy on StrategyQA, which is better than all LUMOS agents that used GPT-3.5-turbo as QA.

- Mind2Web baselines were not fine-tuned at all, hence comparing them to LUMOS is not that fair.

I also have concerns about the significance of this contribution. React and Self-Instruct defined ways to improve the performance of LLM agents with little to no fine-tuning, only publicly available API calls. The concept that large neural networks can be fine-tuned with additional task-specific training data for better performance on those tasks is fairly well-known in the community.

In terms of clarity, the paper has a lot of typos, odd sentences and style issues that made it much more difficult to understand than it should have been. Although they ultimately did not affect the score of this paper, they were close to it. I will list some of the found issues below:

In the abstract: showd -> showed

Introduction

2nd paragraph: However, Lie et al… citations ideally should not be used as nouns.

Beginning of 3rd paragraph: There should be an introductory sentence to ask the question. Then instead of “to this end”” write to answer this question”

In Figure 1, the prompt should be included in the Figure.

End of 3rd paragraph: What are environment states? They are never defined.

4th paragraph: “[...] language agents to acquire these skills [...]” what skills?

2.2

2nd paragraph: “[...] part of grounding module’s input” -> the missing before grounding

2.2 and 2.3 general comment: the difference between LUMOS-I and LUMOS-O should be demonstrated with the same example for easier comprehension (e.g. with the Obama example)

Figure 3a right image:

<|user|> Should we keep planning?
<|assistant|> No I will keep planning.
Shouldn’t the user say “Should we stop planning?”

3.2

3rd paragraph

The subsequent conversation is constructed in the similar patterns. -> remove the

“We assume ourselves as user” -> as the user

“Tell the execution results” -> provide the execution results

“To planning module” -> to the planning module

4th paragraph

“[...] play a user role to provide the task” -> provide to what? Do you mean get/acquire/obtain?

“In the rest conversations” -> remaining, of rest of the

4.4

“Achieves 5-10 average reward than” -> odd sentence

“Than using Self-Instruct method” -> missing the

“Annotation is

 beneficial than” -> more beneficial

Related Work

“We notice that directly generating annotation with for training planning and grounding modules may introduce large number of errors” -> “We notice that directly generating annotations for the training, planning and grounding modules may introduce a large number of errors”

“LLMs transform the gold reasoning steps” -> I don’t understand what you mean, but I am fairly certain not what is written here

**Questions:**

The Self-Instruct paper lists a lot more related work. How are those related to the work presented in this paper?

In the Introduction this sentence is written: “Together, our proposed framework and unified model provides valuable resource and direction for advancing open-source interactive language agents.” (resource -> resources)

But it is never elaborated upon and you do not mention potential future work in the conclusion either.
What future work do you envision after this paper? How could the community successfully build on top of this new framework that was proposed?

What are the limitations of the work you suggested?

---

> ### Author Response · Authors · 2023-11-20
> **Response to Reviewer 7VK9 (1)**
>
> Dear Reviewer 7VK9,
>
> Thank you for engaging with our work and for noting the quality of our methodology and experimental setup. Below we answer your questions:
>
> ---
>
> ### **1. Clarifications about evaluation metrics**
>
> We followed the evaluation metrics adopted in the well-known agent evaluation papers such as AgentBench: a holistic agent evaluation benchmark [1], ReAct [2], AgentLM [3], FiReAct [4], and ReWOO [5]. F1 is actually not the mainstream metric to evaluate agents on complex interactive tasks.
>
> For more comprehensive evaluation as the reviewer suggested, we leverage character-level F1 score, one QA evaluation metric used in ReWOO which evaluates the extent of character overlap, to evaluate agent performance on HotpotQA. Here’re the results:
>
> |             | Base Model    | QA Tool       | Character F1 |
> |-------------|---------------|---------------|--------------|
> | GPT-3.5-CoT | gpt-3.5-turbo | gpt-3.5-turbo | 30.8         |
> | ReAct       | gpt-3.5-turbo | gpt-3.5-turbo | 39.6         |
> | ReWoo       | gpt-3.5-turbo | gpt-3.5-turbo | 40.1         |
> | Lumos-I     | LLAMA-2-7B    | gpt-3.5-turbo | 38.7         |
>
> Via the new metric character-level F1, we find that our LLAMA-2-7B-based Lumos still achieves competitive performance against gpt-3.5-turbo-based agents such as ReAct and ReWOO. It also surpasses GPT-3.5-CoT by a large margin.
>
> [1] AgentBench: Evaluating LLMs as Agents. Liu et al., 2023. Arxiv 2308.03688.
>
> [2] ReAct: Synergizing Reasoning and Acting in Language Models. Yao et al., 2023. ICLR 2023.
>
> [3] AgentTuning: Enabling Generalized Agent Abilities for LLMs. Zeng et al., 2023. Arxiv 2310.12823.
>
> [4] FireAct: Toward Language Agent Fine-tuning. Chen et al., 2023. Arxiv 2309.05653.
>
> [5] ReWOO: Decoupling Reasoning from Observations for Efficient Augmented Language Models. Xu et al., 2023. Arxiv 2305.18323.
>
> ---
>
> ### **2. Reason why we don’t study publicly available APIs**
>
> GPT-based APIs can be excessively costly, especially for tasks involving lengthy contexts like web agent tasks, where encoding HTMLs is a necessity. Also, these APIs are seldom deterministic, making agent reproduction challenging. Moreover, the closed-source nature of these LLMs hinders a comprehensive understanding of their architectures and internal behaviors.
>
> Moreover, commercial use constraints may force companies to develop their proprietary agents based on open-source models. Such reliance on closed-source LLM-based agents is not conducive to the growth of the research and industry community, suggesting a shift towards the use of open-source LLMs.
>
> Lastly, instead of simply showing the effect of fine-tuning, we discuss which training formulations could further improve the agent fine-tuning performance and which data construction methods could contribute to better training annotations for planning and grounding modules. Lumos is one of the first works that study the key elements in agent training process in detail.
>
> ---
>
> ### **3. Clarification about the adopted results in Table 1**
>
> - “On GSM8K, ReWOO achieves 62.4% accuracy”: According to the first sentence of Section 3.2.1 in [ReWOO paper](https://arxiv.org/pdf/2305.18323.pdf), “Table 2 shows the main evaluation results on public benchmarks and curated dataset based on gpt-3.5-turbo”, 62.4% indicates gpt-3.5-turbo results.
> - “Cannot find the claimed LLAMA-7B results in Xu et al. 2023”: Figure 6 in the ReWOO paper demonstrates their LLAMA-7B-based Planner results.
> - “Achieves 66.6% accuracy on StrategyQA”: The 66.6% accuracy is also from Table 2 in the ReWOO paper, which is about gpt-3.5-turbo result.
>
> ---
>
> ### **4. Comparisons on Mind2Web**
>
> In a recent paper [AgentTuning](https://arxiv.org/abs/2310.12823) published (10/19) after ICLR submission deadline, they fine-tune LLAMA-2-70B with Mind2Web training data. 7B-size Lumos (27.7% step success rate) still exceeds AgentLM-70B (13.5% step success rate) by a large margin. We will append the new comparison results and discussion in Section 4.2.

---

> ### Author Response · Authors · 2023-11-20
> **Response to Reviewer 7VK9 (2)**
>
> ### **5. Typo fixing**
>
> Thanks for pointing out the typos! We will address these editorial comments, proofread, and improve the writing of the paper in the revision.
>
> ---
>
> ### **6. Other questions**
>
> **6.1. Main commonality between the related works in Self-Instruct paper and the ones in ours**:
>
> Self-Instruct mainly discusses the method of improving LLMs with their self-generated instruction tuning data. One of the main contributions of Self-Instruct is the automatic approach to distilling GPT-3 for generating instruction-tuning data for further training. As Lumos constructs the training annotations with the aid from GPT-4, one of the common parts of both paper’s related works involve “language models for data generation” which is the second paragraph in Section 5 of Self-Instruct paper.
>
> Also, the common usage of Self-Instruct in recent literatures such as Alpaca [6] and Symbolic Chain-of-Thought Distillation [7] is to synthesize annotations from scratch for training small LMs. Since the goal of Lumos is to equip small LMs with agent ability, one intuitive solution is distilling agent tuning data from LLMs for training. Hence, another common part of the related works is about “knowledge distillation” which is the fifth paragraph in the Self-Instruct paper.
>
> [6] Stanford Alpaca: An Instruction-following LLaMA model. Taori et al., 2023. https://crfm.stanford.edu/2023/03/13/alpaca.html
>
> [7] Symbolic Chain-of-Thought Distillation: Small Models Can Also" Think" Step-by-Step. Li et al., 2023. ACL 2023.
>
> **6.2. Main difference between the related works in Self-Instruct paper and the ones in ours**:
>
> Self-Instruct focuses on relatively simple instruction following settings, e.g., studying whether LLMs can write an email to a professor. However, Lumos focuses on evaluating if language agents can accomplish the complex interactive tasks that require multi-step reasoning with interaction with external environments.
>
> Thus, we discuss language agents which are not mentioned in Self-Instruct. Besides, the instructions generated by Self-Instruct are usually much simpler than our studied complex interactive tasks. Relying simply on Self-Instruct to create complex interactive task annotations can lead to a significant number of errors and diminish the overall quality of the annotations.
>
> In Lumos “Improving Capacity of Smaller Models” paragraph, we further discuss the motivation of transferring the existing annotated gold reasoning steps into the formats aligning Lumos formulations, instead of directly applying the Self-Instruct method.
>
> **6.3. Impact and future works**
>
> As we mentioned in the previous rebuttal part, closed-source API-based agents could suffer from issues regarding affordability, reproducibility, and intellectual property. Pushing forward the development of building strong open-source language agents would be a more proper solution that considers the capabilities and alleviates the aforementioned issues.
>
> Lumos is one of the very first works comprehensively studying which training formulations and data sources should be incorporated to facilitate the performance on complex interactive tasks. In terms of the long-term future works, we hope that based on Lumos training formulations and high-quality training annotations, the future language agent research could shed light on designing more effective, efficient and controllable methods to obtain powerful open-source language agents.
>
> With regard to short-term goals, we plan to train the next-generation Lumos with self-reflection ability. In other words, the next-generation Lumos should realize the planning mistakes they have made and automatically correct the previous planned subgoals until the mistake disappears. Also, we aim to harmonize the language agents with reinforcement learning that makes the planning more adaptive to the external environment and further optimizes the planning modules.
>
> **6.4. Limitations**
>
> Currently, Lumos is unable to replan and reground once the grounded actions cannot be executed. A more flexible agent framework with self-correction ability is desired. Besides, when testing the agents based on Lumos-Iterative (Lumos-I) formulation, we sometimes encounter a situation where the agent cannot terminate the planning at the proper time. A controllable stopping mechanism for the iterative agent formulation is also expected.
>
> ---
>
> ### **Thank you!**
>
> Thank you for your insightful questions and valuable advice. If you have any more questions, we're more than willing to continue the discussion. If you find that our response addresses your concerns, could you please consider increasing your rating score for our paper? Your consideration is highly appreciated.

---

> > ### Author Response · Authors · 2023-11-21
> > **Gentle Reminder**
> >
> > Dear Reviewer,
> >
> > We hope you are doing well. As the discussion period concludes tomorrow (Nov. 22nd), we wanted to reach out to see if you have any follow-up questions. If so, we would appreciate the opportunity to respond before the discussion period ends. We believe our above messages should have addressed your concerns, and therefore may warrant an increase in score if you find them helpful as well. Would you please let us know if you have any further questions or concerns? We are happy to continue the discussion.
> >
> > Thank you very much again for your thoughtful review and help in improving the paper. We appreciate your time and consideration.
> >
> > Best regards,
> >
> > Authors of Lumos

---

> > > ### Author Response · Authors · 2023-11-22
> > > **Kindly Reminder (2)**
> > >
> > > Dear Reviewer,
> > >
> > > We hope you are doing well. As the discussion deadline is approaching, we wanted to know if you have any follow-up questions. If so, we would appreciate the opportunity to respond before the discussion period ends! If our above messages address your concerns, would you like to kindly raise your score? We appreciate your time and consideration.
> > >
> > > Last day, one reviewer raised 2 scores. Also, in our new general response, we further show the better generalizability of Lumos on a new unseen task, InterCode-SQL. Please feel free to let us know if you have further questions!
> > >
> > > Best regards,
> > >
> > > Authors of Lumos

---

> > > > ### Comment · Reviewer_7VK9 · 2023-11-23
> > > >
> > > > Dear Authors,
> > > >
> > > > Thank you for your detailed response. Unfortunately these last two days of the rebuttal period have been very busy for me. I will thoroughly read and evaluate your response and update my score according to it later this week.

---

> > > > > ### Author Response · Authors · 2023-11-23
> > > > > **Take your time and Good luck!!**
> > > > >
> > > > > Dear Reviewer 7VK9,
> > > > >
> > > > > No worries. We totally understand this. Please take your time! Also, best wishes to your rebuttal!
> > > > >
> > > > > Best,
> > > > >
> > > > > Authors of Lumos

---

### Official Review · Reviewer_9Mw2 · 2023-11-01

**Soundness:** 2 fair
**Presentation:** 3 good
**Contribution:** 2 fair
**Rating:** 5
**Confidence:** 4

**Summary:**

The paper presents LUMOS, a language agent framework built for open source LLMs with unified formats and modular design. LUMOS divide the framework into separate modules for planning, grounding and execution. The obtain high-quality annotations for training the modules, the authors leverage LLMs to convert ground truth intermediate reasoning steps in existing benchmarks into a unified format. LUMOS demonstrates competitive or superior performance compared to SOTA systems on a variety of interactions including web agent, math reasoning and complex QA.

**Strengths:**

1. The paper studies an important problem of developing language agents, and a unified framework and format is much needed in the field.
2. The overall description of the method is clear and easy to follow.
3. Experiment covers both regular fine-tuning setting, and generalization to unseen task.

**Weaknesses:**

1. My main concern is regarding the claim on LUMOS achieving superior performance than other LLM-based agents:
    1. LUMOS is trained on top of LLAMA-2, while some of the baselines are based on LLAMA. For example in Table 1, according to the latest results from AgentBench, updated vicuna-13B v1.5 based on LLAMA-2 now has 41.7 on WebShop, even outperform LUMOS. To make a fair comparison, I would recommend to keep it consistent across models, reporting both LUMOS and baselines with LLAMA, or update the baseline results to the version using LLAMA-2. This should also be made more clear in the paper.
    2. In most experiments, LUMOS is tuned with data from downstream tasks, while if I understand correctly other LLM baselines are tested under few-shot or in-context learning settings. If this is correct, it should be made more clear in the paper, and the comparison seem a little unfair.
2. With the baseline systems mostly evaluated under few-shot, I feel it is important to understand the efficiency of LUMOS and how well it generalizes. It is great that the authors have the generalization on WebShop, but I feel more emphasis on this direction, with additional few-shot experiments would be much better.
3. A modular design, in particular divide the agent into planning, grounding and execution has been studied in various previous works as well. This limits the novelty of the proposed method. Also there is some missing reference to relevant work, e.g., Saycan: Grounding Language in Robotic Affordances
4. Many of the baseline results are directly taken from results reported in other papers. While I understand that running experiments with LLMs are costly, this causes some in-consistency in the baselines that are compared in different datasets. And the comparison might also get affected by the implementation details of different papers.

**Questions:**

1. Is it true that most other baseline LLM agents are applied under few-shot / in-context learning setting?

---

> ### Author Response · Authors · 2023-11-20
> **Response to Reviewer 9Mw2 (1)**
>
> Dear Reviewer 9Mw2,
>
> We thank the reviewer for engaging with our work, and recognizing the importance of the unified framework we contribute in this paper.
>
> **_We would like to address your concerns in detail below._**
>
> ---
>
> ### **1. Better baseline model comparison**
>
> Prior to delving into our recent comparative experiments, it's essential to provide context and a timeline concerning the agent evaluation and modeling studies.
>
> As of the ICLR submission deadline (9/28), there existed only ONE public version of AgentBench [1], which was released on 8/7. All the results referenced in Lumos are derived from this initial version. The subsequent version, referred to in the review, was released later on 10/25. Therefore, we are not able to cite results from Vicuna-13B v1.5 based on LLAMA-2 from AgentBench Version 2 [2] prior to the ICLR deadline. We will update it in the following versions.
>
> [1] (Version 1) AgentBench: Evaluating LLMs as Agents. Liu et al., 2023. https://arxiv.org/abs/2308.03688v1
>
> [2] (Version 2) AgentBench: Evaluating LLMs as Agents. Liu et al., 2023. https://arxiv.org/abs/2308.03688v2
>
> Additionally, prior to 9/28, the dominant method in language agent research up to that date was few-shot in-context learning, exemplified by models like ReAct, ReWOO, and Reflexion. Very few studies had explored the fine-tuning of smaller LMs to compete with the performance of GPT-series or other significant open language agents.
>
> Consequently, we did not specifically emphasize in-context learning settings at that time. After the submission deadline, novel approaches to language agent fine-tuning began to emerge, including FiReAct (10/9) and AgentLM (10/19), marking the beginning of research into enhancing smaller LMs with language agent capabilities through fine-tuning.
>
> In terms of the comparison with LLAMA-2-based agents, we note that some LLAMA-2-based agent performance is already shown in Table 1 and we have already provided some comparison. To further address the reviewer’s concern, we compare 1) more baselines of which the base models belong to LLAMA-2 series, and 2) the concurrent two language agent fine-tuning methods, FiReAct [3] and AgentLM [4].
>
> [3] FireAct: Toward Language Agent Fine-tuning. Chen et al., 2023. Arxiv 2309.05653.
>
> [4] AgentTuning: Enabling Generalized Agent Abilities for LLMs. Zeng et al., 2023. Arxiv 2310.12823.
>
> ---
>
> ###  **2. Lumos vs. other LLAMA-2-based models**
>
> - For math tasks, some baselines listed in Table 1(b) are based on LLAMA-2. Described in Table 3 and 4 of [MAmmoTH](https://arxiv.org/abs/2309.05653) [5], the base model of Orca-Platypus-13B baseline is LLAMA-2. It is trained with many math CoT data constructed upon FLAN Collection. 7B-size Lumos-O outperforms Orca-Platypus-13B on GSM8K and SVAMP 12.1% and 8.6% accuracy.
> - For web agent task, Mind2Web, shown in Table 3 of the latest [AgentBench v2](https://arxiv.org/abs/2308.03688), Vicuna-13B v1.5 only gets a 12% step success rate, ~15% underperforming Lumos. LLAMA-2-70B-chat also lags behind Lumos by a large margin.
> - For WebShop, also shown in Table 3 of the latest [AgentBench v2](https://arxiv.org/abs/2308.03688), Lumos outperforms LLAMA-2-13B-chat 14.5 average reward.
>
> [5] MAmmoTH: Building Math Generalist Models through Hybrid Instruction Tuning. Yue et al., 2023. Arxiv 2309.05653.
>
> ---
>
> ### **3. Lumos vs. FiReAct and AgentLM**
>
> - For math tasks, Lumos achieves 18.1% accuracy improvements over AgentLM-13B on GSM8K (based on Table 4 of AgentBench).
> - For complex QA tasks, we use Exact Match (EM) as the evaluation metric to keep it consistent with FiReAct and AgentLM. Despite FiReAct being fine-tuned with in-domain HotpotQA annotations, Lumos, without any fine-tuning on HotpotQA annotations, still presents an impressive improvement. In particular, Lumos surpasses the 7B-scale FiReAct by 3.2% EM. Lumos also has a 7.1% EM improvement over AgentLM.
> - For web agent tasks, Mind2Web, Lumos surpasses AgentLM-70B fine-tuned on Mind2Web 14.1% step success rate.
>
> Overall, we show that Lumos is still competitive with many LLAMA-2-based agent baselines and has superior performance to the concurrent works on most of our evaluated tasks. We will update our paper with these new results.

---

> ### Author Response · Authors · 2023-11-20
> **Response to Reviewer 9Mw2 (2)**
>
> ### **4. More few-shot generalization experiments**
>
> We extend the few-shot generalization experiments on WebShop to 13B-scale. Specifically, We adopt LLAMA-2-13B as the fundamental model for both the planning and grounding modules. We show the results as follows:
>
> |                    | Reward |
> |--------------------|--------|
> | Vicuna-v1.3-33B    | 23.9   |
> | Vicuna-v1.5-13B    | 41.7   |
> | Openchat-v3.2-13B  | 46.9   |
> | Claude-instant     | 49.7   |
> | Lumos-I-math       | 45.7   |
> | Lumos-I-complex-QA | 47.3   |
> | Lumos-I-web-agent  | 46.2   |
> | Lumos-I            | 50.3   |
>
> Experimental results suggest that 1) Lumos still has better generalizability than the other LLAMA-2-13B-based models even API-agent agents under the same few-shot in-context learning settings.; 2) Lumos-I trained with unified annotations also outperforms the ones trained with domain-specific annotations Lumos-I-math/complex-QA/web-agent on unseen tasks.
>
> ---
>
> ### **5. Novelty on model architecture**
>
> Instead of simply presenting a plan-ground-execute modular architecture, we focus more on studying 1) how to **train** a better open-source language agent, and 2) how to prepare high-quality planning and grounding **annotations** that align with the proposed formulations.
>
> In terms of the training formulations, since there are very few prior works exploring agent fine-tuning, it is unclear which formulations would be more beneficial to train a better agent. In fact, many well-known agent frameworks (e.g., ReAct and ReWOO) fully rely on a single model to perform planning and grounding. However, we are unsure about whether training a single model is the most appropriate approach to managing planning and grounding skills.
>
> Additionally, it is also possible that chain-of-thought fine-tuning is already good to tackle math and complex QA tasks [6]. As shown in Section 4.3, we unveil the effect of possible open-source agent training formulations. It could provide a practical reference to lead the path for training more powerful open-source agents in the future.
>
> Regarding the training annotation construction, one intuitive solution is to simply leverage methods such as Self-Instruct to generate tasks, plans and grounded actions by LLMs from scratch. Whereas, these methods are not suitable for generating high-quality annotations for complex interactive tasks, as even GPT-4 performs badly on the complex interactive tasks we study, such as Mind2Web. Relying on these methods to create complex interactive task annotations can lead to a significant number of errors and diminish the overall quality of the annotations. To this end, it is necessary to think of potential novel methods that can create **large-scale and high-quality** annotations to train planning and grounding modules.
>
> Finally, we propose a prompting-based conversion method that utilizes the ground-truth reasoning steps in existing benchmarks to generate the subgoals and actions and transfer to conversational formats that fit the proposed formulations. Appendix E demonstrates that by controlling the training annotation size to be the same, our annotations can still help get better performance than the ones produced by the Self-Instruct method and passed by rigorous execution sanity checking.
>
> Overall, Lumos involves the discussion beyond merely presenting an architecture. We care more about what training strategies can construct a better open-source agent. It covers the comprehensive study on the choices of the key elements in the agent training process including training formulation and annotation perspectives.
>
> [6] Distilling Reasoning Capabilities into Smaller Language Models. Shridhar et al., 2023. ACL Findings 2023.

---

> > ### Comment · Reviewer_9Mw2 · 2023-11-22
> >
> > Thanks for the detailed response and additional experiments. They addressed some of my concerns on fair comparison with baselines. Again, my concern is not on that the paper does not achieve SOTA, or missing comparison with concurrent works, but rather a fair and consistent comparison across models, if most baselines are using LLAMA, then a LLAMA based model is better suited for comparison, with LLAMA-2 based models to further demonstrating that the proposed method can generalize.
> > I will raise my score to weak reject. My remaining concern is on generalization of the method. While I appreciate the efforts on consolidating existing supervised datasets and the study on fine-tuning, **large-scale and high-quality** data is not easy to get in many cases, especially for agent applications that involve complex interaction and environment. The experiment on generalization to Webshop touches on this but seems a bit insufficient with only one dataset and even with all the fine-tuning seems many chat model are still competitive with the proposed method, somehow hint that general instruction tuning might be suffice.

---

> > > ### Author Response · Authors · 2023-11-22
> > > **Response to Reviewer 9Mw2's New Questions**
> > >
> > > Dear Reviewer 9Mw2,
> > >
> > > Thank you so much for raising the score! Below we answer your new questions:
> > >
> > > ---
> > >
> > > ### **1. Additional generalization experiments**
> > >
> > > We test the generalizability of Lumos on a new unseen task called InterCode-SQL. InterCode-SQL is a benchmark for evaluating agents on the interactive coding task. Given a natural language instruction, an agent is required to interact with a database with code to resolve the issue. Due to the time limit, we select 200 testing samples from InterCode-SQL and perform evaluation on them. The experiments are based on 13B-size Lumos agents. Here’re the success rate of each agent (%):
> > >
> > > |                    | InterCode-SQL |
> > > |--------------------|---------------|
> > > | Vicuna-v1.5-13B    | 4.0           |
> > > | Vicuna-v1.3-13B    | 6.0           |
> > > | Vicuna-v1.3-33B    | 6.5           |
> > > | Lumos-I-maths      | 5.5           |
> > > | Lumos-I-complex-qa | 3.0           |
> > > | Lumos-I-web-agent  | 4.5           |
> > > | Lumos-I            | **7.0**       |
> > >
> > > We show that Lumos-I outperforms larger agents such as Vicuna-v1.3-33B. It also has superior performance to the domain-specific agents and Vicuna-v1.5 based on the same base model LLAMA-2-13B.
> > >
> > > ---
> > >
> > > ### **2. Getting more diverse agent training data is feasible**
> > >
> > > In fact, there exist multiple complex interactive tasks containing the ground-truth intermediate reasoning steps and encompassing various domains, such as AlfWorld (embodied house-hold tasks), ScienceWorld and WebShop (virtual text game tasks), NL2Bash (interactive coding task), etc. The number of the gold annotations is not trivial and allows further expansion of our training annotations. We argue that our contribution is a framework for training agents and prove the advantages of Lumos, so we are confident that more data collection and transformation can be done to further improve the generalization and performance of Lumos.
> > >
> > > ---
> > >
> > > ### **3. Necessity of agent fine-tuning**
> > >
> > > We believe that using our unified data format to collect diverse training data can fine-tune agents that can do better cross-task generalization, as we’ve shown in the WebShop and InterCode experiments. For unseen tasks or emerging environments, fine-tuned agents with more diverse training data will have likely performed better for generalization.
> > >
> > > In addition, we refer to the results in AgentTuning (Zeng et al., 2023, a concurrent work published after ICLR submission deadline). On 6 unseen agent tasks, agent fine-tuning brings a relative 76%, 57% and 176% overall performance improvement (measured by AgentBench metric), in comparison with the 7/13/70B-scale LLAMA-2-chat models trained with general instruction tuning methods. This finding is aligned with ours.
> > >
> > > ---
> > >
> > > We sincerely thank you for your constructive suggestions and questions. If our new experiments and evidence address your concerns, would you kindly consider raising your rating score again? We greatly appreciate your consideration!
> > >
> > > Authors of Lumos

---

> ### Author Response · Authors · 2023-11-20
> **Response to Reviewer 9Mw2 (3)**
>
> ### **6. Clarification about baseline model comparison**
>
> To address the reviewer’s concern about baseline result comparison in Table 1, we conduct the prompting experiments for each task type’s strongest baselines of which the calling prompts are publicly released in AgentBench and ReWOO [7] papers. Especially for WebShop, we adopt the prompts described in AgentBench paper and rewrite their in-context examples to be consistent with ours. The results are shown as follows:
>
> |                 | Eval. Dataset | Perf. | Lumos-I Perf. |
> |-----------------|---------------|-------|---------------|
> | Alpaca-7B       | GSM8K         | 38.3  | 46.7          |
> | ReWOO           | StrategyQA    | 65.7  | 65.7          |
> | ReWOO           | HotpotQA      | 44.2  | 45.9          |
> | Vicuna-v1.1-13B | WebShop       | 15.7  | 39.8          |
> | GPT-4           | Mind2Web      | 26.8  | 27.7          |
>
> We show that Lumos-I still exceeds the compared baselines in Table 1 after the reproduction.
>
> [7] ReWOO: Decoupling Reasoning from Observations for Efficient Augmented Language Models. Xu et al., 2023. Arxiv 2305.18323.
>
> —
>
> ### **7. Other questions**
>
> “Is it true that most other baseline LLM agents are applied under few-shot / in-context learning setting?” - Yes, we will add this information in the latter version. Recently, two fine-tuned language agents, FiReAct (10/9) and AgentTuning (10/19) were published after the ICLR submission deadline (9/28). As discussed in the rebuttal subsection **Lumos vs. FiReAct and AgentLM**,  Lumos outperforms them on most of our evaluation tasks.
>
> —
>
> ### **Thank you!**
> ​
> We appreciate your excellent questions and suggestions. Please feel free to reach out if you have additional questions. If you find that our response addresses your concerns, would you kindly consider raising your rating score for our paper? We greatly appreciate your consideration.

---

> > ### Author Response · Authors · 2023-11-21
> > **Gentle Reminder**
> >
> > Dear Reviewer,
> >
> > We hope you are doing well. As the discussion period concludes tomorrow (Nov. 22nd), we wanted to reach out to see if you have any follow-up questions. If so, we would appreciate the opportunity to respond before the discussion period ends. We believe our above messages should have addressed your concerns, and therefore may warrant an increase in score if you find them helpful as well. Would you please let us know if you have any further questions or concerns? We are happy to continue the discussion.
> >
> > Thank you very much again for your thoughtful review and help in improving the paper. We appreciate your time and consideration.
> >
> > Best regards,
> >
> > Authors of Lumos

---

### Official Review · Reviewer_UN8U · 2023-11-01

**Soundness:** 3 good
**Presentation:** 3 good
**Contribution:** 3 good
**Rating:** 8
**Confidence:** 4

**Summary:**

The paper proposes LUMOS, Language agents with Unified formats, Modular design, and Open Source LLMs to solve complex tasks with planning, grounding, and execution modules fine-tuned from LLAMA-7B on high-quality annotations collected by leveraging LLMs to convert ground truth reasoning steps in existing benchmarks into a unified format. LUMOS achieves competitive performance with agents of larger size and outperforms GPT-4/3.5-based agents on complex QA and web agent tasks.

**Strengths:**

- The proposed modular framework is well-motivated.

- The converted dataset can contribute to training better small open models for complex tasks.

- The results show the proposed method is effective and promising for its generalizability on unseen tasks.

- The paper is well-written and presented clearly.

**Weaknesses:**

- Why is LUMOS-O better than LUMOS-I on Math benchmarks?

- Some discussion on the performance-efficiency tradeoff between LUMOS-O and LUMOS-I would provide further insights.

- Figure 3 (a) in the Final planning module annotation "**No**, I will keep planning. Subgoal 2: Query the living
period of Jonathan Kaplan." Is that a typo?

- Some related work on modular language agents framework for complex tasks
  - _Cognitive Architectures for Language Agents_
  - _Building Cooperative Embodied Agents Modularly with Large Language Models_

**Questions:**

See weaknesses.

---

> ### Author Response · Authors · 2023-11-20
> **Response to Reviewer UN8U**
>
> Dear Reviewer UN8U,
>
> Thank you for your thoughtful review! We are glad you found the modular framework to be well-motivated and noticed the potential of our dataset to help train better, smaller models. Thank you also for noting the effectiveness and generalizability of our method. Below we answer your questions:
>
> ---
>
> ### **1. Reason why Lumos-O is better than Lumos-I on math benchmarks**
>
> A potential reason why Lumos-I is not better than Lumos-O on math tasks is because the calculation results of the intermediate reasoning subgoals planned by Lumos-I are not useful for generating the next subgoal. Suppose we are solving a math problem like “James decides to run 3 sprints 3 times a week. He runs 60 meters each sprint. How many total meters does he run a week?” Even if the agent calculates how many times James sprints, which is 3*3=9, the mere number 9 does not affect the next subgoal generation, which could be “Calculate the total meters James runs a week”.
>
> ---
>
> ### **2. Performance-efficiency tradeoff between Lumos-O and Lumos-I**
>
> We compute the inference time for Lumos-O and Lumos-I across 100 instances on GSM8K and HotpotQA, respectively. The experiments are run with 2 NVIDIA A6000 48GB GPUs with inference batch size 16. We find that Lumos-O is much more efficient than Lumos-I on both datasets.
>
> |         | GSM8K | HotpotQA |
> |---------|-------|----------|
> | Lumos-O | 102 s | 556 s    |
> | Lumos-I | 851 s | 1007 s   |
>
> Lumos-O completes its inference in a single round, whereas Lumos-I necessitates multiple rounds of inference until it autonomously concludes its planning. The iterative planning and grounding in Lumos-I contribute to a higher time cost for solving individual instances. Nevertheless, Lumos-I is better at generating an appropriate subgoal based on the current external environment compared to Lumos-O.
>
> For example, when tackling a complex question “What government position was held by the woman who portrayed Corliss Archer in the film Kiss and Tell?”, Lumos-I is able to first identify the woman who portrayed Corliss Archer in Kiss and Tell, Shirley Temple, then asks the government position she held.
>
> However, though Lumos-O is able to first ask who the woman is, without the interaction with external knowledge such as Wikipedia pages, it will generate a subgoal which inquires the government position held by `Madonna`, a random entity totally irrelevant with the original question. Hence, Lumos-O is a more efficient solution, but not as effective as Lumos-I due to the lack of the adaptability to external environments.
>
> ---
>
> ### **3. Typo in final planning module annotation**
>
> Sorry about this typo. The previous turn in this conversation actually asks “Should we stop planning?”, instead of “Should we keep planning?” We will fix the typo in the camera-ready version.
>
> ---
>
> ### **Thank you!**
> ​
> Thank you very much for your great questions and suggestions. Please let us know if you have any further questions, as we are happy to continue the discussion. If you find that our response addresses your concerns, would you kindly consider defending acceptance for our paper or even raising your rating score? We greatly appreciate your consideration.

---

> > ### Author Response · Authors · 2023-11-21
> > **Gentle Reminder**
> >
> > Dear Reviewer,
> >
> > We hope you are doing well. As the discussion period concludes tomorrow (Nov. 22nd), we wanted to reach out to see if you have any follow-up questions. If so, we would appreciate the opportunity to respond before the discussion period ends. We believe our above messages should have addressed your concerns, and therefore may warrant an increase in score if you find them helpful as well. Would you please let us know if you have any further questions or concerns? We are happy to continue the discussion.
> >
> > Thank you very much again for your thoughtful review and help in improving the paper. We appreciate your time and consideration.
> >
> > Best regards,
> >
> > Authors of Lumos

---

> > ### Comment · Reviewer_UN8U · 2023-11-23
> >
> > Thank you for your reply. My concern is addressed.

---

### Author Response · Authors · 2023-11-21
**General Response (1)**

Dear Reviewers,

We are thankful for your thoughtful reviews and pleased that you have recognized the strengths of our paper. We briefly summarize them as follows:
​
- **Well-motivated, solid and impactful framework**: “a unified framework and format is much needed in the field”, “achieve better results by prompting LLMs in a nascent field, which could be seen as original”, etc.
- **Better performance than larger agents and detailed experiments**: “detailed ablations of the method, making it stronger”, “effective and promising for its generalizability on unseen tasks”, etc.
- **Great writing and easy-to-follow method**: “clear and easy to follow”, “well-written and presented clearly”, “images are mostly clearly captioned”, etc.
​
In our rebuttal, we would like to emphasize our efforts in the following aspects:

---
​
### **1. Rigorous baseline model comparison**

We first clarify that as of the ICLR submission deadline (9/28), there existed only ONE public version of AgentBench [1], which was released on 8/7. All the results referenced in Lumos are derived from this initial version. We are unable to cite the latest LLAMA-2-based results provided in AgentBench V2 (published on 10/25) such as Vicuna-v1.5 in our ICLR submission. However, we already incorporated some comparison between the baselines based on LLAMA-2 (e.g., Lumos vs. Orca-Platypus-13B).

For fairer and more comprehensive comparison, in our rebuttal, we conduct compare Lumos with 1) more baselines that leverage LLAMA-2 series base models, and 2) the concurrent two agent fine-tuning methods, `FiReAct` [2] (published on 10/9) and `AgentLM` [3] (published on 10/19). We show that Lumos still performs better than many LLAMA-2-based agent baselines and has superior performance to the concurrent works on most of our evaluated tasks. Specifically, despite FiReAct being fine-tuned with in-domain HotpotQA annotations, Lumos, without any fine-tuning on HotpotQA annotations, surpasses the 7B-scale FiReAct by 3.2% EM. Lumos also has a 7.1% EM improvement over AgentLM. Lumos also outperforms AgentLM-70B fine-tuned on Mind2Web 14.1% step success rate. We will update these new results in the future version.

We also reproduce some prompting experiments shown in Table 1 under the fairly similar evaluation settings to ours by controlling the in-context examples to be consistent. We show that Lumos-I still exceeds the compared baselines in Table 1 after the reproduction.

[1] AgentBench: Evaluating LLMs as Agents. Liu et al., 2023. Arxiv 2308.03688.

[2] FireAct: Toward Language Agent Fine-tuning. Chen et al., 2023. Arxiv 2309.05653.

[3] AgentTuning: Enabling Generalized Agent Abilities for LLMs. Zeng et al., 2023. Arxiv 2310.12823.

---
​
### **2. Motivation and impact of proposing Lumos**

GPT-based APIs can result in high calling costs, particularly for tasks with long contexts such as web agent tasks that require HTML encoding. Additionally, these APIs pose challenges for agent reproduction. The closed-source nature of these APIs further impedes the understanding of their architectures and internal behavior, compromising their reliability. Lastly, limitations on commercial use may compel companies to create their own proprietary agents using open-source models. All of these highlight the drawbacks of depending on closed-source LLM-based agents and suggest a move towards developing open-source LLM-based agents for the benefit of the research and industry community.

As one of the first studies focusing on how to train a better open-source language agent, we hope to provide valuable agent training resources and benchmarks, and lead the direction for advancing more effective and reliable language agents.

---

> ### Author Response · Authors · 2023-11-21
> **General Response (2)**
>
> ### **3. Novelty of Lumos architecture**
> ​
> Instead of simply presenting a plan-ground-execute modular architecture, we focus more on studying 1) how to **train** a better open-source language agent (Section 2, 3 and 4.3), and 2) how to prepare high-quality planning and grounding **annotations** that align with the proposed formulations (Section 3 and Appendix E).
>
> Concretely, in terms of training formulations, we discuss the detailed input and output contents of each module, how to build the collaboration between the modules, where the supervision signals could be constructed, and comparison with multiple potential agent training formulations.
>
> Regarding annotation construction, we propose a prompting-based conversion method to acquire high-quality large-scale annotations that align with the formulations of planning and grounding modules. We conduct rigorous experiments to show the quality of our annotation construction method with a comparison to the commonly used Self-Instruct method by controlling the number of annotations.
>
> To our best knowledge, Lumos is one of the first studies concentrating on investigating the detailed components (training formulations and data) involved in agent training strategies prior to ICLR submission deadline.
>
> ---
>
> ### **Thank you!**
> ​
> We sincerely thank the reviewers for their constructive suggestions and questions to enhance our paper. Please reply if you have any further questions, and we will be more than happy to continue the discussion.

---

### Author Response · Authors · 2023-11-22
**General Response (3)**

### **Additional generalization experiments**

We test the generalizability of Lumos on a new unseen task called InterCode-SQL. InterCode-SQL is a benchmark for agents on the interactive coding task. Given a natural language request, an agent is asked to interact with a database with code to resolve the issue. Due to the time limit, we select 200 testing samples from InterCode-SQL and perform evaluation on them. The experiments are based on 13B-size Lumos agents. Here’re the success rate of each agent (%):

|                    | InterCode-SQL |
|--------------------|---------------|
| Vicuna-v1.5-13B    | 4.0           |
| Vicuna-v1.3-13B    | 6.0           |
| Vicuna-v1.3-33B    | 6.5           |
| Lumos-I-maths      | 5.5           |
| Lumos-I-complex-qa | 3.0           |
| Lumos-I-web-agent  | 4.5           |
| Lumos-I            | **7.0**       |

We show that Lumos-I outperforms larger agents such as Vicuna-v1.3-33B. It also has superior performance to the domain-specific agents and Vicuna-v1.5 based on the same base model LLAMA-2-13B.

---

### Meta-Review · Area_Chair_5Ldb · 2023-12-06

**Metareview:**

This paper introduces LUMOS, a modular and open-source framework for language agents built upon open-sourced LLMs. LUMOS leverages a three-stage pipeline with dedicated modules fine-tuned from open-sourced LLMs for planning, grounding, and execution to tackle complex tasks like math reasoning, QA, and web navigation. By utilizing high-quality annotations generated by GPT-4, LUMOS achieves competitive or even superior performance compared to larger proprietary models (including GPT-4/3.5) while remaining computationally efficient. In general, the reviewers appreciate the motivation, strong empirical results, and solid experiments/ablation. Most of the concerns have been addressed during rebuttal, e.g. unfairness in comparison to prior art. However, the AC wants to echo a reviewer opinion on the difficulties of acquiring large-scale, high-quality, and diverse agent experiences for fine-tuning. The AC agrees with the authors' point that general instruction-tuning is likely not sufficient, but it's highly unclear how to do it at scale -- this can largely limit the value of the proposed framework. Thus the AC recommends rejecting this paper.

In addition, an evaluation that the AC would like to see is how much fine-tuning degrades the performance on other language tasks. It may not be necessary when applying to in-domain tasks but could imply its generalizability. Note that this has no weight in the AC's decision.

**Justification For Why Not Higher Score:**

Limited value of this open framework as it's generally very hard to effectively collect agent experiences at scale for most of the domains.

**Justification For Why Not Lower Score:**

N/A

---

### Decision · Program_Chairs · 2024-01-16

Reject